# Methods of Granulocyte Isolation from Human Blood and Labeling with Multimodal Superparamagnetic Iron Oxide Nanoparticles

**DOI:** 10.3390/molecules25040765

**Published:** 2020-02-11

**Authors:** Fernando Alvieri, Javier B. Mamani, Mariana P. Nucci, Fernando A. Oliveira, Igor S. Filgueiras, Gabriel N. A. Rego, Marycel F. de Barboza, Helio R. da Silva, Lionel F. Gamarra

**Affiliations:** 1Hospital Israelita Albert Einstein, São Paulo 05652-900, Brazil; fernando.alvieri@einstein.br (F.A.); javierbm@einstein.br (J.B.M.); fernando.anselmo@einstein.br (F.A.O.); igor.filgueiras@usp.br (I.S.F.); gnery.biomedicina@gmail.com (G.N.A.R.); marycel.barboza@einstein.br (M.F.d.B.); herodriguessilva@gmail.com (H.R.d.S.); 2LIM44—Hospital das Clinicas HCFMUSP, Faculdade de Medicina, Universidade de São Paulo, São Paulo 01246-903, Brazil; marinucci@usp.br

**Keywords:** granulocyte isolation, magnetic nanoparticle, SPION, magnetic resonance imaging, near-infrared fluorescence, ICP-MS, molecular imaging, inflammation, infection

## Abstract

This in vitro study aimed to find the best method of granulocyte isolation for subsequent labeling with multimodal nanoparticles (magnetic and fluorescent properties) to enable detection by optical and magnetic resonance imaging (MRI) techniques. The granulocytes were obtained from venous blood samples from 12 healthy volunteers. To achieve high purity and yield, four different methods of granulocyte isolation were evaluated. The isolated granulocytes were labeled with multimodal superparamagnetic iron oxide nanoparticles (M-SPIONs) coated with dextran, and the iron load was evaluated qualitatively and quantitatively by MRI, near-infrared fluorescence (NIRF) and inductively coupled plasma mass spectrometry (ICP-MS). The best method of granulocyte isolation was Percoll with Ficoll, which showed 95.92% purity and 94% viability. After labeling with M-SPIONs, the granulocytes showed 98.0% purity with a yield of 3.5 × 10^6^ cells/mL and more than 98.6% viability. The iron-loading value in the labeled granulocytes, as obtained by MRI, was 6.40 ± 0.18 pg/cell. Similar values were found with the ICP-MS and NIRF imaging techniques. Therefore, our study shows that it is possible to isolate granulocytes with high purity and yield and labeling with M-SPIONs provides a high internalized iron load and low toxicity to cells. Therefore, these M-SPION-labeled granulocytes could be a promising candidate for future use in inflammation/infection detection by optical and MRI techniques.

## 1. Introduction

Inflammation is the immune system′s response to pathological agents (infection), damaged cells and toxins, and it aims to maintain integrity, defense against microorganisms and cellular and tissue repair [1,2,3]. Acute inflammation of the lower respiratory tract is a leading cause of mortality and morbidity worldwide, with the greatest prevalence in children younger than five years [4,5,6]. Sepsis is another important public health problem; it occurs in 30% to 50% of hospitalizations that culminate in death [7,8] due to exacerbated and inappropriate inflammation, and it can cause septic shock and organ failure [9]. In addition, the chronic inflammation process increases therapy and care costs, as well as mortality [10].

Hard-to-reach or occult inflammation and infection processes are clinically challenging. Their clinical diagnoses involve biochemical and radiological examinations, but these methods can produce false negatives [11,12,13,14,15]. To determine the localization, extent and severity of inflammation, imaging techniques such as computed tomography (CT) and magnetic resonance imaging (MRI) can be performed but have limitations in the early stages of the disease, as they are dependent on morphological changes [16,17,18]. Accurate and early diagnosis of inflammation and infection helps reduce mortality and morbidity and increase drug treatment success [19,20,21].

Proinflammatory effects help leukocytes migrate from the blood vessel to the site of injury, due to increased vascular permeability through stimulation by cytokines and interleukins. There is an initial accumulation of granulocytes, mainly neutrophils, in the inflamed area and later of lymphocytes and macrophages [22,23,24]. Granulocytes are the most abundant subset of leukocytes [25] and are produced in the bone marrow, originating from myeloid precursor cells and undergoing a maturing process before they are able to perform their functions, mainly phagocytosis [25,26]. They differ from other cells in that they have granules with proteolytic enzymes that fight microorganisms or other inflammatory agents [25,27]. Granulocytes are considered frontline cells in inflammation and infection processes [28]. In many cases, to detect inflammation/infection, it is necessary to label isolated granulocytes with contrast agents. Granulocytes can be isolated from whole blood by different methods. The most commonly used method uses gradients of Percoll [29,30,31,32,33,34]. Isolation purity, viability and yield are important factors for evaluating each method and ensuring cellular integrity [29,34,35,36].

The noninvasive evaluation of acute inflammatory processes makes it possible to analyze molecular and cellular processes in vivo, usually based on targeting specialized cells such as macrophages, lymphocytes or granulocytes to the desired target [37]. Noninvasive imaging techniques that provide structural, functional and molecular information in preclinical and clinical studies include CT, MRI, positron emission tomography (PET), single-photon emission computed tomography (SPETC) and near-infrared fluorescence (NIRF) imaging, among others [38,39,40], each of which has advantages and limitations. The MRI technique offers high spatial resolution but low sensitivity, high cost and long scanning time. The PET technique provides high sensitivity and excellent penetration depth but incurs high cyclotron production costs and radiation risk. The SPECT technique has high sensitivity, excellent penetration depth and low radiation risk [41]. The NIRF technique has high resolution with specific molecular contrast but limited penetration. Thus, a combination of imaging techniques (multimodal imaging) can overcome the disadvantages of each and provide complementary information on the inflammation/infection process. The accurate detection of various pathologies requires multimodal agents that have optical, magnetic, fluorescent and radioactive properties, among others, with at least two such properties in a single probe [38,42,43] that have adequate retention in the place of interest, that present contrast differences between healthy and pathological tissues and that have low cost and easy preparation, all resulting in a multimodal image containing anatomical, functional and metabolic information [16,17,44].

One of the possibilities for contrast agents is nanoparticles, which have the potential to be used by various molecular imaging techniques due to their range of magnetic, fluorescent and radiative properties, among others [45]. The most commonly used nanoparticles are metals, Qdots, liposomes, lipid nanoparticles, micelle nanotubes, quantum dots, dendrimers, fluorescent nanoparticles and polymeric nanoparticles [46]. The specific characteristics of nanoparticles, such as size, surface charge, morphology, type of functionalized or nonfunctionalized biocompatible polymer coating, coupling of visible or infrared fluorophores and coupling of radioisotopes, can influence the labeling and detection of granulocytes by molecular imaging techniques [47].

Superparamagnetic iron oxide nanoparticles (SPIONs) are outstanding MRI contrast agents due to their excellent magnetic properties and low toxicity, which allow the use of special surface coatings with organic or inorganic molecules, including surfactants, drugs, proteins, starches, enzymes, antibodies, nucleotides, nonionic detergents and polyelectrolytes [48], in addition to being approved for clinical use [49]. There are several strategies for detecting inflammatory processes, one of which is to label autologous immune cells exogenously with SPIONs and inject them systemically for homing by noninvasive imaging techniques [50,51,52,53,54]. Another method is the endogenous labeling of the monocyte/macrophage lineage using ultrasmall superparamagnetic iron oxide (USPIO) or microparticle iron oxide (MPIO) [50,55,56].

Most studies employing exogenously labeled granulocytes to analyze infection or inflammation processes have used SPIONs with different strategies [51,54,57,58,59,60,61,62]. SPIONs, when coupled with near-infrared spectrum fluorophores, enhance pathological assessments due to the sensitivity or specificity of the complementary information provided by the optical imaging modality [63], enabling multimodal imaging [64,65].

Thus, this in vitro study aimed to find the best (highest purity and yield) granulocyte isolation technique, with the aim of labeling the granulocytes with multimodal nanoparticles detectable by MRI-NIRF imaging, for qualitative and quantitative evaluations of their internalized M-SPION loads, a promising candidate for future research use in detecting inflammation/infection processes by optical and MRI techniques.

## 2. Results

### 2.1. Evaluation of Granulocyte Isolation by Different Methods

Granulocytes were isolated by four different methods: method I—Percoll gradients, method II—Percoll with phosphate buffer solution (PBS), method III—Percoll with Hank′s balanced salt solution (HBSS) and method IV—Percoll and Ficoll. Then, we assessed the purity of the granulocytes isolated by each method (Table 1). The highest percentage mean purity was found in method IV, at 95.92% (Figure 1A-iv), in which the granulocytes were deposited in the pellet, followed by method III, at 82.75% (Figure 1A-iii), in which the granulocytes were concentrated in the white layer between the Percoll densities. Method I had a purity of 57.89% (Figure 1A-i), and the granulocytes were concentrated in the white layer above the sediment of the red blood. Method II had a purity of 47.86% (Figure 1A-ii), and all cells sedimented without visible separation. These results are depicted in the boxplot (Figure 1B). The nonparametric ANOVA test (Kruskal-Walls test) showed a significant difference between methods (*p* < 0.001). The post hoc test showed a significant difference in all comparisons (*p* < 0.008; Table 2).

In the same samples used for the granulocyte purity analysis, a contaminant analysis was performed for each method (Table 3). The lowest percentage mean contamination was found in method IV (0.88%) (Figure 1A-iv), followed by method III (10.95%) (Figure 1A-iii), method I (20.67%) (Figure 1A-i) and method II (37.85%) (Figure 1A-ii). No granulocyte isolation was observed in method II, due to the unique pellet formation. These results are depicted in the boxplot (Figure 1C). The nonparametric ANOVA test (Kruskal-Walls test) showed a significant difference between methods (*p* < 0.001). The post hoc test showed a significant difference in all comparisons (*p* < 0.003; Table 4).

### 2.2. Characterization, Morphology, Viability and Yield of the Methods with High Purity

Figure 2 compares some properties of the granulocytes isolated by methods III and IV. Panels A1 and B1 present forward-scatter versus side-scatter plots of the granulocyte isolation rates of method III (30.6%) and method IV (58.1%), respectively. Panels A2 and B2 show the respective proportions of isolated granulocytes that were double-positive for CD15/CD13. Panels A3 and B3 show the percentages of leukocytes that were positive for CD45 (98.5% and 99.5%, respectively). Panels A4 and B4 show the percentages of granulocytes that were double-negative for CD19/CD14 (94.3% and 97.6%, respectively). Panels A5 and B5 show the percentages of granulocytes that were double-negative for CD3/CD14 (86.5% and 98%, respectively).

In the fast panoptic staining, we detected the percentages of granulocytes in both methods (yellow arrows in Figure 2C3 and 2D3). The isolated granulocytes made up a mean of 6.3% of cells (eosinophils, basophils and immature neutrophils) in method III (blue arrows in Figure 2C3) and 3.2% in method IV (blue arrows in Figure 2D2). They made up 20.68% of lymphocytes and contaminants in method III (red arrows in Figure 2C3) and only 0.61% of lymphocytes and contaminants in method IV.

The granulocyte viability was analyzed after 4 h of isolation through membrane integrity by flow cytometry, which showed 98.3% of granulocytes isolated by method III were viable (Figure 2E) and 94.0% of those isolated by method IV (Figure 2F). Both were double-negative for Annexin-FITC and propidium iodine-phycoerythrin (PI-PE).

The purity analysis of isolated granulocytes and their respective contaminants in methods III and IV showed a proportional inverse relationship, in which method IV had the highest and the lowest contaminant rate, as depicted in the boxplot of Figure 2 (G and H). The purity (Table 5) and contaminant percentage (Table 6) were significantly different (*p* < 0.001) between methods III and IV. In relation to the isolated granulocyte yield, means of (4.99 ± 1.24) × 10^6^ per mL and (3.51 ± 1.40) × 10^6^ per mL were found by methods III and IV, respectively, without a significant difference between methods, as shown by a Student′s *t*-test (*p* = 0.996). The isolated lymphocyte yield between methods III and IV were significantly different by the Mann-Whitney test (*p* = 0.025), with respective means of (0.33 ± 0.07) × 10^6^ per mL and (0.09 ± 0.06) × 10^6^ per mL.

### 2.3. M-SPION Optical Characterization and Analysis of Size Polydispersion, Stability, and Zeta Potential

The M-SPION optical characterization depicted in Figure 3 shows the excitation/emission spectrum of M-SPIONs. The spectrum shows the visible fluorescent spectrum peak (ex/em: 559.4/580.7 nm) and NIRF absorption/emission (ex/em: 757.0/777.4 nm).

Figure 3B shows the polydispersion curve adjusted to a log-normal distribution with an average hydrodynamic diameter of 35.7 ± 0.3 nm, compatible with the manufacturer’s specifications. For M-SPION stability evaluation, the polydispersion curves were acquired temporally until reaching 20 h. In this period, the M-SPION hydrodynamic diameter did not change, and there was no evidence of M-SPION agglomeration due to the interaction of M-SPIONs with the elements that contained culture medium.

To analyze the electrostatic interaction of granulocytes with M-SPIONs, the surface charge was measured using the same pH of 7.4 found in the cell labeling, and a zeta potential of approximately 32 mV was verified.

### 2.4. Visualization of M-SPION Internalization into Granulocytes and Cellular Viability

After labeling the isolated granulocytes with M-SPIONs, internalization was verified through fluorescence (Figure 4A,C,E–G) and brightfield microscopy (Figure 4B,D). Figure 4A shows the labeling of granulocytes with M-SPIONs by fluorescence microscopy, in which the fluorescent rhodamine-coupled M-SPIONs are highlighted in red, and the labeling of granulocyte nuclei by 4′,6-diamidine-2′-phenylindole dihydrochloride (DAPI) is shown in blue. An amplified view (inset of Figure 4A) allowed us to better visualize the presence of M-SPIONs internalized into granulocytes. In brightfield microscopy analysis, it was possible to visualize the granulocytes labeled with M-SPIONs while highlighting the iron of the intracellular region of granulocytes by Prussian blue staining, which better visualized the labeling of granulocytes in the inset of Figure 4B. Figure 4C,D show images corresponding to the same field of view of the granulocytes labeled with M-SPIONs by fluorescence and brightfield microscopy. Therefore, the magnetic and fluorescent properties of these nanoparticles allowed detection by both techniques.

Figure 4E–G show the M-SPION internalization in each part. First, rhodamine was imaged by fluorescence microscopy (Figure 4E), followed by the image corresponding to granulocyte nuclear staining by DAPI (Figure 4F) and overlapping images (Figure 4G), where M-SPIONs were observed in the granulocyte cytoplasm.

After 4 h of labeling, granulocyte viability was evaluated with M-SPION concentrations of 10, 30 and 50 µg Fe/mL, which showed viability (double-negativity for Annexin-FITC and PI-PE) values of 99.3%, 99.6% and 98.6%, respectively (Figure 4H–J).

### 2.5. Quantification of M-SPIONs Internalized into Granulocytes by MRI, ICP-MS and NIRF

MRI, ICP-MS, and NIRF were used to quantify the iron mass internalized into granulocytes after M-SPION incubation at concentrations of 10, 30 and 50 µg Fe/mL. The quantification results of the iron mass per cell and the number of nanoparticles per granulocyte are shown in Table 7 and Figure 5.

#### 2.5.1. MRI Quantification

The MRI quantification was begun by calculating the r2 value; for this purpose, the T2-weighted image of the phantom was obtained using different echo times (TE) and containing the following concentrations of M-SPIONs dispersed in 1% agarose at: 5, 10, 15, 20, 25, 30, 35, 40 and 50 µg Fe/mL (Figure 5A). The contrast decay of well images was evidenced by the M-SPION concentration rise for a set time of TE. From the signal intensity of the MRI (in arbitrary units (AU)), the transverse relaxation curves were obtained as a function of TE (ms), as depicted in Figure 5A. The curves showed exponential decay behavior that varied with the M-SPION concentration, and the corresponding T2 parameters were obtained from the exponential adjustment. To obtain the r2 value, a curve of the inverse transverse relaxation time (1/T2) was constructed as a function of iron concentrations, and then a linear adjustment was obtained for the relaxivity value of r2 = (19.9 ± 0.9) × 10^−4^ ms^−1^ µg Fe^−1^ mL for M-SPIONs, as depicted in the inset of Figure 5A.

For the iron load calculation of Equation (1), the T2 values of granulocytes labeled with M-SPION concentrations of 10, 30 and 50 µg Fe/mL and of the control granulocytes not labeled with M-SPIONs were calculated. From the curves of the signal intensity of MRI as a function of TE (Figure 5B), the following T2 values were obtained: 56.6 ± 2.1 ms for the control and 47.5 ± 1.2 ms, 36.8 ± 1.9 ms and 23.2 ± 1.1 ms for the M-SPION concentrations of 10, 30 and 50 µg Fe/mL, respectively, as shown in the boxplot in the inset of Figure 5B. After determining the r2 and T2 values, it was possible to quantify the iron load in pg Fe/cell and the number of M-SPIONs/cell by MRI using Equations (1) and (2). The iron load quantified by MRI for the M-SPION concentrations 10, 30 and 50 µg Fe/mL was 0.83 ± 0.11 pg/cell or (1.04 ± 0.14) × 10^4^ M-SPION/cell, 2.41 ± 0.15 pg/cell or (3.03 ± 0.19) × 10^4^ M-SPION/cell and 6.40 ± 0.18 pg/cell or (8.06 ± 0.22) × 10^4^ M-SPION/cell, respectively (Figure 5C and Table 7). Figure 5C shows the contrast of the MR images produced by M-SPIONs internalized into granulocytes.

#### 2.5.2. NIRF Quantification

After construction of the NIRF calibration curve with known concentrations of M-SPIONs and the signal intensity of NIRF, the relation [NIRF Intensity] = [(1.86 ± 0.07) × [Fe] + (1.89 ± 0.14)] × 10^8^ was obtained, as shown in Figure 5D. From the calibration curve, the iron load values (pg Fe/cell and M-SPIONs/cell) of the different SPION concentrations internalized into granulocytes were obtained. For the M-SPION concentration of 10 µg Fe/mL, the iron load was 0.59 ± 0.10 pg/cell or (0.74 ± 0.13) × 10^4^ M-SPIONs/cell; for the M-SPION concentration of 30 µg Fe/mL, it was 3.49 ± 0.19 pg/cell or (4.39 ± 0.23) × 10^4^ M-SPIONs/cell and for the M-SPION concentration of 50 µg Fe/mL, it was 4.64 ± 0.14 pg/cell or (5.84 ± 0.18) × 10^4^ M-SPIONs/cell (Figure 5E and Table 7).

#### 2.5.3. ICP-MS Quantification

The calibration curve for the ICP-MS quantification is shown in Figure 5F, in which the values of the net experimental intensity (cps) as a function of the iron concentration (ppb) were plotted, and the linear adjustment yielded the relation [Fe] = (−580.0 ± 51.8) + (982.9 ± 37.7) × [net intensity]. From the calibration curve, the following values of iron load internalized into granulocytes were obtained: 1.08 ± 0.26 pg/cell or (1.35 ± 0.32) × 10^4^ M-SPIONs for the concentration of 10 µg Fe/mL, 2.56 ± 0.16 pg/cell or (3.21 ± 0.20) × 10^4^ M-SPIONs at 30 µg Fe/mL and 6.83 ± 0.47 pg/cell or (8.56 ± 0.59) × 10^4^ M-SPIONs at 50 µg Fe/mL (Figure 5G and Table 7).

The differences between the iron load values found in the MRI, NIRF and ICP-MS quantifications of the M-SPIONs internalized into granulocytes are shown in the radar graph of Figure 5H.

## 3. Discussion

Granulocytes are known for their high potential and efficiency in detecting and eradicating microbial infections through an immune response against invading pathogens [66]. Aiming at granulocyte-labeling with nanoparticles for later use in inflammation/infection process detection, different methods of granulocyte isolation were tested, and their labeling with multimodal nanoparticles allowed their internalization and their qualitative and quantitative evaluations.

Among the methods of granulocyte isolation evaluated, the method that achieved the highest degree of purity was the use of Percoll gradients, followed by the Ficoll method (method IV). However, until now, no method using the combination of Percoll and Ficoll has been reported, only each technique separately. In addition, there is no consensus on the best technique for isolating granulocytes to achieve high purity. The methods applied for granulocyte isolation are diverse [29,30,31,33,34,35,53,67,68,69,70], and their efficiency has been reported in different ways, such as cellular viability or cell isolation yield, cell functionality, number of target cells and contaminants after isolation, which makes it difficult to evaluate the methods for clinical application.

Method IV, which resulted in the highest purity of granulocyte isolation, was performed in two stages. The first stage consisted of the use of Percoll gradients with HBSS (method III), and the second stage used Ficoll on the granulocytes isolated by method III. This first stage of granulocyte isolation was used in some studies as a unique stage of isolation [29,31,32], and the purity analysis, granulocyte yield and cellular viability were reported as 81% purity in 4 × 10^6^ cells/mL [31], 95% purity in 2.2 × 10^6^ cells/mL [29] and 95% purity for the yield range of 1 to 2 × 10^6^ cells/mL, as well as 95% cellular viability [32]. These results corroborate our findings of 86.5% purity in 5.0 × 10^6^ cells/mL and 98.3% cellular viability. After the second stage of isolation with Ficoll (method IV), which yielded 98.0% purity, a 3.5 × 10^6^ cells/mL isolation yield and 94.0% cellular viability were found. The studies that used only Ficoll for granulocyte isolation reported purity greater than 90% but with a low isolation yield, ranging from 0.6 to 1.0 × 10^6^ cells/mL [51,53,68,71,72].

Considering these results, after the first stage of Percoll (method III), we were able to isolate more granulocytes than described in the literature. In addition to superior performance in terms of high viability, we found similar purity to those described in other studies, which used the same method of isolation with Percoll gradients with HBSS [29,31,32]. After the use of Ficoll (method IV), the granulocyte yield decreased by 30% but without a significant difference compared to method III, maintaining 94% cellular viability and 98% purity in the granulocyte isolation. The reduction in the number of contaminants was significant, especially regarding the number of lymphocytes present, which was reduced by almost 73% after method IV. The goal of this study was to isolate specific cells without contaminants, especially for clinical applications, since the cell viability remained adequate for application.

The influences of Percoll and Ficoll on the functional integrity of granulocytes after isolation have been evaluated by different studies [72,73,74], which have demonstrated unaltered phagocytic features through the analysis of the percentage and number of particles phagocytized.

For granulocyte labeling, nanoparticles must have appropriate physicochemical properties in relation to size, coating, zeta potential, concentration and stability; these proprieties contribute to efficient internalization, maintaining the cell viability of the granulocytes [75,76,77]. Efficient labeling with the contrast agent is important in granulocyte detection by different imaging techniques supporting various clinical applications, such as immune response, diagnosis and therapies [51,78,79,80,81].

Regarding the nanoparticle property analysis during granulocyte labeling, the use of M-SPIONs with a 35 nm diameter yielded a zeta potential of + 32 mV [76], with adequate temporal stability of the M-SPIONs, and coating with dextran allowed adequate internalization (~85%) with high cellular viability (>98.6%). In addition, a high-intensity signal was detected by MRI, NIRF and brightfield/fluorescence microscopy at the nanoparticle concentrations used, and the labeling time of 4 h has been considered adequate for clinical applications [61].

For internalization, it is best for the nanoparticles to have a positive surface charge (zeta potential), given the negative surface charge of granulocytes [75,82] and the electrostatic process involved [77,83,84]. Therefore, granulocyte and leukocyte-labeling studies that have used SPIONs with a negative zeta potential (0 to −53 mV) [51,54,57,58,85,86] have reported the necessity of using chemical or physical transfection agents or other strategies that would contribute to SPION internalization into cells and boost the internalization yield [51,54,57,86].

Other relevant aspects of labeling are the incubation time of cells with SPIONs, the SPION concentration used and its corresponding viability. In our study, we found satisfactory viability values using a 4-h labeling time (99.3%, 99.6% and 98.6% for the respective M-SPION concentrations of 10, 30 and 50 μg Fe/mL). In studies on granulocyte and leukocyte-labeling with SPIONs, the incubation time was reported to be between 1 and 4 h for labeling [51,54,57,59,61,62,85,86]. For example, in the study of Tang et al. [61], an incubation time of 3 h was best for the clinical application of these cells. However, other studies [53,58,60] reported SPION incubation times up to 24 h. Previous studies have reported that SPION concentrations between 1 and 200 μg Fe/mL showed cellular viability greater than 86% [51,57,58,59,60,85,87,88], whereas cellular viability was reduced at concentrations from 5 to 20 mg Fe/mL [53,54,86]. The dextran coating used in M-SPIONs has been reported as adequate for biomedical applications in terms of safety and biocompatibility [89] and is used in most granulocyte-labeling studies to cover nanoparticles [51,57,59,60,61,88].

Regarding the stability of nanoparticles in the labeling process, previous studies by our group [90,91,92] evaluated SPION stability before labeling in different media and demonstrated that supplementation with 10% fetal bovine serum (FBS) contributed stabilized SPIONs in the culture medium during the labeling process [90,91,93]. This type of strategy was also reported in other studies [54,58,59,85,87], with FBS concentrations ranging from 5% to 20% during the labeling process of granulocytes or leucocytes with SPIONs. The use of FBS helps avoid the nanoparticle agglomerations that can occur due to the interaction of SPIONs with a culture medium that is supplemented with proteins and electrolytes [91,93], as well as helping maintain the equilibrium of existing forces involved in the interaction between the nanoparticles, such as electrostatic forces, van der Waals forces, steric forces and magnetic forces modulated by the Brownian motion associated with nanoparticles [93].

In most studies, SPION internalization into granulocytes or leukocytes is confirmed using Prussian blue staining and MRI [51,54,57,61], because the composition of nanoparticles is based on iron oxide. However, as the nanoparticles used in our study have multimodal features (magnetic/dual fluorescence), it was possible to verify internalization using Prussian blue staining and MRI, as well as using fluorescence images in the visible spectrum (rhodamine) and infrared spectrum (IR750). The evaluation by brightfield and fluorescence microscopy showed that most granulocytes were labeled with M-SPIONs, corroborating other studies that reported similar results by brightfield microscopy evaluation [51,54], as the nanoparticles used in these studies only had iron oxide nuclei. The unlabeled granulocytes may be related to their maturation, because, in earlier stages of maturation, there are descriptions of the incomplete functionality of granulocytes for the phagocytosis process [94]. The granulocyte stages of maturation were not evaluated in our study, which can be considered one of the study limitations and an explanation of the unlabeled granulocytes. The isolation of granulocyte populations from each stage of granulocyte maturation present in venous blood and their marking could help to better understand this result.

This study lays the foundation for future applications of granulocyte labeling with nanoparticles for inflammation/infection process detection by in vivo imaging techniques. The nanoparticles used (M-SPIONs) can be detected by MRI and NIRF techniques. The MRI qualitative analysis showed that the hypointense signal of the granulocytes labeled at M-SPION concentrations of 10, 30 and 50 µg Fe/Ml, when compared with the control image (unlabeled granulocytes), had a signal reduction of 13%, 37% and 74%, respectively, compared to a TE of 48.6 ms and a repetition time (TR) of 3000 ms. The corresponding T2 values were 56.6 ms for the control, 47.5 ms (labeling with 10 µg Fe/mL), 36.8 ms (labeling with 30 µg Fe/mL) and 23.2 ms (labeling with 50 µg Fe/mL). These results show adequate internalization and, due to the M-SPIONs, a high r2/r1 ratio of 165.8 (r2 = (19.9 ± 0.9) × 10^−4^ ms^−1^ µg Fe^−1^ mL and r1 = (1.2 ± 0.5) × 10^−5^ ms^−1^ µg Fe^−1^ mL), indicating a strong contrast in T2-weighted images [95,96,97,98,99]. In relation to the NIRF qualitative analysis, the NIRF intensity signal was possible due to the M-SPIONs used for granulocyte-labeling having a fluorophore coupled to the infrared wavelength. The NIRF intensity signal of the lowest M-SPION concentration was on the order of 10^8^ photons/s (labeling with 10 µg Fe/mL), and, at 30 and 50 µg Fe/mL, the SPION concentrations were six and eight times higher than the lowest concentration. Thus, it was possible to obtain MRI and NIRF signals in in vivo studies.

The quantitative analysis of the iron load internalized into granulocytes using the MRI, ICP-MS and NIRF techniques shows values (Table 7) that are in accordance with the signal detection limitations as a function of sensitivity, spatial resolution, temporal resolution and the concentration of contrast agent [38,39]. In studies by Shanhua et al. [54,86], quantification analysis of the iron load of the SPIONs internalized into granulocytes was performed using the ICP technique, which showed an iron load of 0.1265 pg/cell (0.1265 ppm/106 cell) in the best strategy of internalization using LPS and 5 mg/mL SPIONs. Comparing these results with our quantitative analysis by the same technique revealed a much higher iron charge using a lower M-SPION concentration. With M-SPION concentrations of 10, 30 and 50 μg Fe/mL, the iron loads were 1.08, 2.56 and 6.83 pg/cell, respectively. Thus, the iron load internalized into cells in our study using the lowest M-SPION concentration was 10 times higher than the best condition of Shanhua et al. [54,86], and the highest M-SPION concentration yielded a result 54 times higher. The quantifications by MRI and NIRF techniques showed similar results, as depicted in Table 7. When the granulocyte viability after labeling was calculated, the highest M-SPION concentration used in our study showed 98% viability, whereas the studies that used 100 times more SPIONs reported 70% cellular viability after labeling [54,86].

One of the limitations of SPION exogenous-labeling of immune cells is the low sensitivity of MRI detection [50] compared to endogenous-labeling (MPIO) techniques, where MRI detection sensitivity is higher and provides high contrast [56,100] but has the disadvantage of false positive results in cases where hemorrhages and blood oxygen level-dependent (BOLD) effects induce signal voids in T2*-weighted images, as well as regarding toxicity, lack of biodegradability and accumulation in the reticuloendothelial system, which limit clinical use [50]. To improve the low sensitivity of exogenous-labeling used in our study, (i) the efficiency of isolation purification was purified, which is related to the increased yield of isolated cells, as well as their purity and (ii) the use of immunological cell-labeling strategies to increase internalized SPION load without affecting cell viability, which resulted in higher purity and with higher cell isolation yield than those obtained in the literature [29,31,32,51,53,68,71,72], as well as greater SPION load internalization compared to the literature [54,86,101], thus, increasing the SPION detection sensitivity by MRI techniques and their viability in future clinical applications. SPION detection by MRI using exogenous-labeling already reported in in vivo studies [101,102,103,104] and Krieg’s study [101] showed that the quantification of the internalized SPION load on immune cells in the in vitro experiment was lower than that found in our study, using a 500-fold lower concentration in the labeling process, although even with a lower internalized SPION load, it was possible to detect the SPION signal (hypo-intensity image) by MRI in the in vivo study. Therefore, the M-SPIONs used point to a promising candidate for in vivo studies.

Therefore, our study showed that it is possible to isolate granulocytes with high purity and yield, and the labeling with M-SPIONs provided a high internalized iron load and low toxicity to cells. These multimodal nanoparticles with magnetic and fluorescent features are detectable in both the visible spectrum of fluorescence and the infrared spectrum and can be used in in vitro, ex vivo and in vivo studies for inflammation/infection process detection using various imaging techniques, such as MRI and NIRF. For these images, the acquisition of a single probe (M-SPION) is necessary, providing complementary information to the pathology diagnosis, considering the limitations of each technique. In addition, this study highlights a strong future for the in vivo investigation of inflammation/infection using multimodal nanoparticle-labeled granulocytes.

## 4. Materials and Methods

### 4.1. Granulocyte Isolation from Human Peripheral Blood

Granulocytes were obtained from healthy volunteers′ blood samples, and all subjects gave their informed consent for inclusion before participation in the study. The study was conducted in accordance with the Declaration of Helsinki, and the protocol was approved by the Ethics Committee of the Albert Einstein Hospital (CAAE—66749517.0.0000.0071).

From each volunteer (*n* = 12), 20 mL of peripheral venous blood was collected using a 20-G needle to avoid lysing cells, followed by the addition of 50 UI heparin sodium. In granulocyte isolation, the experimental design compared 4 different methods: Percoll density gradients, Percoll density gradients with PBS, Percoll density gradients with Hanks′ solution and Percoll gradients and Ficoll. Each method is described in detail below, and Figure 6 represents the schematic differences. The granulocyte isolation methods were compared to find the one with the highest purity.

### 4.2. Granulocyte Isolation Methods

#### 4.2.1. Isolation method with Percoll Density Gradient

First, the total leukocytes were separated from the venous blood, and 5 mL of voluven (Fresenius Kabi, Brazil) was added to the blood sample to increase the hemosedimentation speed and allowed to stand for 45 min. After this time, the leukocytes were now visible as a white layer between the red cells and plasma (buffy coat). The leukocytes were collected to begin the process of granulocyte isolation with Percoll density gradients, as described by de Vries et al. [67] and Roca et al. [69].

Two different densities of Percoll gradients (P4937, Sigma Aldrich, São Paulo, Brazil) were used for granulocyte isolation. Gradient solution 1 (G1, Figure 6) with 56% Percoll consisted of 1.0 mL of 9% NaCl, 5.5 mL of 100% Percoll and 3.4 mL of water for injection. Gradient solution 2 (G2, Figure 6) with 71% Percoll consisted of 1.0 mL of 9% NaCl, 9.0 mL of 100% Percoll and 1.9 mL of water for injection. The Percoll gradient was performed by slowly adding the following sequence: 3 mL of gradient solution 2 (G2), 3 mL of gradient solution 1 (G1) and 3 mL of the leukocyte mixture gently deposited on top, taking care to avoid mixing. The Percoll gradients were centrifuged at 300× *g* for 30 min at 22 °C without a centrifugation brake. After centrifugation, the supernatant was removed, and the white layer was carefully collected with granulocytes between the gradients and resuspended in 4 mL of DMEM (Sigma Aldrich, São Paulo, Brazil) supplemented with 10% FBS for granulocyte purity analysis, as depicted in Figure 6, method I.

#### 4.2.2. Isolation method with Percoll Density Gradients and PBS

The second granulocyte isolation method followed the same total leukocyte separation process described above (Percoll density gradients). In this method, the Percoll gradients were constructed using different densities by diluting Percoll with PBS (Gibco^®^, Carlsbad, CA, USA). Gradient solution 3 (G3, Figure 6) and gradient solution 4 (G4, Figure 6) contained 60% and 70% Percoll, respectively, as described in previous studies [35,70,105]. The Percoll solutions with PBS were obtained by slowly adding the following sequence: 3 mL of gradient 4 (G4), 3 mL of gradient 3 (G3) and 3 mL of the leukocyte mixture gently deposited on top, taking care to avoid mixing. These gradients were centrifuged at 100× *g* for 20 min at 22 °C without centrifugation braking. After centrifugation, the granulocyte-containing white layer between the gradient solutions was collected carefully and resuspended in 4 mL of DMEM supplemented with 10% FBS for granulocyte purity analysis, as depicted in Figure 6, method II.

#### 4.2.3. Isolation Method with Percoll Density Gradients and Hanks′ Solution

The third method did not require previous leukocyte isolation. The gradients were constructed with Hanks′ balanced salt solution without calcium and magnesium 10× (HBSS, Sigma Aldrich, Brazil) and Percoll, as used in previous studies [32,33,106,107]. Gradient solution 5 (G5, Figure 6) with 60% Percoll consisted of 1.8 mL of Percoll 100%, 300 µL of HBSS and 900 µL of Milli-Q water, and gradient solution 6 (G6, Figure 6) with 70% Percoll consisted of 2.1 mL of Percoll 100%, 300 µL of HBSS and 600 µL of Milli-Q water.

The Percoll gradient with Hanks′ solution was obtained by slowly adding the following sequence: 3 mL of gradient 6 (G6), 3 mL of gradient 5 (G5) and 3 mL of venous blood, gently deposited on top, taking care to avoid mixing. These solutions were centrifuged at 400× *g* for 30 min at 22 °C without centrifugation braking. After centrifugation, the granulocytes were visible as a white layer between gradient 6 and gradient 5. The supernatant was removed, and the granulocyte layer collected was added to 4 mL of 1 × HBSS to recover the cellular osmolarity. Then, another centrifugation was performed at 150× *g* for 5 min at 22 °C without centrifugation braking. The pellet sediment was resuspended in 4 mL of DMEM supplemented with 10% FBS for granulocyte purity analysis, as depicted in Figure 6, method III.

#### 4.2.4. Isolation Method with PERCOLL Gradients and Ficoll

The fourth method consisted of improved granulocyte purification, which combined the result of the granulocyte isolation method (III) using Percoll density gradients with Hanks′ solution and another centrifugation with Ficoll Premium (17544602GE Healthcare, Brazil). Continuing the process described above, 4 mL of granulocytes isolated and suspended in DMEM were added carefully above 4 mL of Ficoll Premium and centrifuged at 400× *g* for 30 min at 22 °C without centrifugation braking. The pellet sediment was resuspended in 4 mL of DMEM supplemented with 10% FBS for granulocyte purity analysis, as depicted in Figure 6, method IV.

### 4.3. Visualization of Granulocyte Isolation Methods

For visual analyses, 50 µL of isolated granulocytes was added to a cytofunnel with a cytocard and a microscope slide and centrifuged using a cytocentrifuge (Cientec, CT2000, Brazil) at 250× *g* for 3 min. After centrifugation, the granulocytes appeared as agglomerated cells in the middle of the microscope slide. The cells were dried at room temperature to perform fast hematology staining (fast panoptic, Interlab, Brazil) and visualized by convectional optical microscopy (Nikon, TiE).

### 4.4. Evaluation of Isolated Granulocyte Purity

First, granulocyte purity after each method was assessed through cell counting using an automated cell counter (Sysmex, Brasil), and the cells from the highest purity methods were characterized by flow cytometry (LSRFortessa, Bdsciencis). Isolated granulocytes were phenotyped by negative markers (CD14, CD19 and CD3) and positive markers (CD45, CD15 and CD13).

For this purpose, 10^6^ isolated cells in 100 µL were used, and the following granulocyte-specific antibodies were used to confirm their identity as white cells and neutrophils: 5 µL of CD3-AlexaFluor 700-A; 2 µL of CD14-PE-Cy7-A; 5 µL of CD19-APC-Cy7-A (BD Biosciences, EUA) to exclude monocytes and dendritic cells, lymphocytes and lymphocytes B, respectively; 8 µL of CD45-PercP-Cy5.5-A; 5 µL CD15-FITC-A and 5 µL of CD13-PE-A (BD Biosciences, EUA).

### 4.5. Granulocyte Viability after Isolation Methods

Granulocyte viability was analyzed 4 h after granulocyte isolation by the highest purity methods using flow cytometry (LSRFortessa, Bdsciencis). This time was considered plausible for granulocyte isolation, labeling and administration in future studies to detect inflammation and infection processes. The flow cytometry evaluation was performed with 10^6^ isolated cells in 100 µL, 5 μL of annexin V-FITC, 5 μL of propidium iodide-phycoerythrin (PI-PE) (Sigma Aldrich, EUA) and 300 μL of lysis buffer (Excellyse Live, Czech Republic). The red blood cells were lysed for 20 min without light contact. These tubes were centrifuged at 500× *g* for 5 min, the supernatant was removed, and the resuspended sediment was analyzed by flow cytometry. Viable granulocytes were defined as annexin V negative and PI-PE negative.

### 4.6. Multimodal Superparamagnetic Iron Oxide Nanoparticles (M-SPIONs)

For granulocyte labeling, we used multimodal superparamagnetic iron oxide nanoparticles (M-SPIONs), with a crystalline-phase magnetic core identified as magnetite, a hydrodynamic size of 35 nm, an 8 nm iron oxide nucleus coated with dextran, a zeta potential of ~ +31 mV and a density of ~1.25 g/cm^3^ (Biopal, Molday ION™), half of which were conjugated with a fluorophore that emitted NIRF absorption/emission wavelengths in the 750/777 nm range and another half with a fluorophore that emitted rhodamine-B absorption/emission wavelengths in the 558/580 nm range.

### 4.7. M-SPION Optical Characterization and Analysis of Size Polydispersion, Stability, and Zeta Potential

The optical characterization of M-SPIONs dispersed in aqueous medium was performed at 50 µg Fe/mL using an RF-6000 spectrofluorophotometer (Shimadzu, Kyoto, Japan) to acquire the spectrum with excitation in the wavelength range from 520 to 800 nm and emission from 550 to 780 nm.

The polydispersion of hydrodynamic diameter (HD), temporal stability of HD and zeta potential of M-SPIONs (50 µg Fe/mL) were measured using dynamic light scattering (DLS) with Zetasizer Nano S equipment (Malvern, Reino Unido). The HD distribution curve was obtained using an angle of 173 degrees with 15 measurements in 5 s, maintaining a constant temperature at 37 °C. M-SPION HD stability analysis was performed in cell culture medium (DMEM +10% FBS) for 20 h. The zeta potential measurements (surface charge) were performed in the pH range from 7 to 9.

### 4.8. Labeling of Granulocytes with M-SPIONs

Isolated granulocytes were labeled with M-SPIONs using 10^6^ cells per well (48-well plate) resuspended in 4 mL of DMEM (Sigma Aldrich, Brazil) with 10% FBS (Sigma Aldrich, Brazil). M-SPIONs were used at concentrations of 10, 30 and 50 μg Fe/mL and incubated for 4 h at 37 °C. After this period, the cells were washed three times with 0.9% saline solution and centrifuged at 300× *g* for 5 min at 22 °C to remove the M-SPIONs that were not internalized by the granulocytes.

### 4.9. Visualization of M-SPIONs Internalized in Granulocytes

The M-SPION internalization was analyzed by brightfield and fluorescence microscopy images due to the magnetic and fluorescence features of nanoparticles. The granulocytes were deposited on a microscope slide using a cytocentrifuge, as described in Section 4.2. Prussian blue staining was performed with 500 µL of a solution containing 5% potassium ferrocyanide (Sigma Aldrich, St Louis, MO, USA) and 5% hydrochloric acid (Merck, Darmstadt, Germany) for 15 min and washed once with deionized water. Then, Nuclear Fast Red staining was performed with a 1% solution (0.02 g of Nuclear Fast Red in 2 mL of deionized water) for 10 min for nuclear counterstaining, and the nuclei were quickly washed once more and analyzed by optical microscopy. Subsequently, fluorescence analysis was performed using diamidino-2-phenylindole (DAPI, Sigma Aldrich) to label the cell nuclei for 10 min, followed by washing once to register the M-SPION fluorescence image using an excitation/emission filter of 530/550 nm. Both image analyses were performed using a Nikon TiE fluorescence microscope (Tokyo, Japan).

### 4.10. Granulocyte Viability Evaluation after M-SPION Labeling

The granulocyte viability was evaluated after 4 h of labeling with M-SPION concentrations of 10, 30 and 50 μg Fe/mL using the same procedures performed in the flow cytometry analysis with annexin V-FITC and propidium iodide described in Section 4.4.

### 4.11. Quantification of M-SPION Internalized in Granulocytes

After granulocyte labeling, the M-SPIONs internalized in the cells were quantified using MRI, inductively coupled plasma mass spectrometry (ICP-MS) and NIRF techniques. The iron load internalized by the granulocytes was calculated in pg/cell units and as the number of M-SPIONs/cell.

#### 4.11.1. Quantification by MRI

The MRI quantification was initiated with relaxometry characterization of M-SPIONs to calculate the transverse relaxivity (r2) value. A phantom with M-SPIONs dispersed in 1 mL of agarose 1% (Sigma Aldrich, St Louis, MO, USA) was prepared for each well at concentrations of 0, 5, 10, 15, 20, 25, 30, 35, 40 and 50 µg Fe/mL. The images were acquired in the PET-MRI hybrid tomography equipment 3T (Trio, SIEMENS, Germany) for the entire body, with a 32-channel head coil using a T2-weighted sequence (multicontrast turbo-spin echo, SE_MS); 31 different echo times (TE = 8, 16, 24,…,256 ms); fixed repetition time (TR = 3000 ms); slice thickness of 3 mm; FOV of 15.9 × 20.0 cm^2^; matrix of 256 × 256 and 16 averages. Then, the phantom images were analyzed by SyngoVia software (Siemens, Germany) using a region of interest (ROI) for each well of the phantom image to extract the MRI intensity signals acquired in the different TEs. These data were adjusted with the exponential decay equation I = Io exp(−TE/T2), where I is the signal intensity and Io is the initial signal intensity, to obtain the T2 relaxation time values. The inverse transverse relaxation rate values (1/T2) as a function of the M-SPION concentration of the phantom wells were adjusted linearly to obtain an r2 that corresponded to the inclination coefficient of a straight line.

For quantification of M-SPIONs internalized into the cell, a phantom for each well containing 2 × 10^6^ granulocytes was dispersed in 1 mL of 1% agarose under the following conditions: unlabeled granulocytes and granulocytes labeled with 10, 30 and 50 µg Fe/mL of M-SPIONs. The MRI data were obtained in a similar way to r2, providing the T2 values for the respective concentrations studied. Finally, the M-SPION captured by granulocytes was determined using Equation (1):(1)$$\frac{1}{T_{2}^{Labeled\;Granulocyte}}=\frac{1}{T_{2}^{not\;Labeled\;Granulocyte}}\;+\;r_{2}*\left[M-SPION\right],$$
where $1/T_{2}$ is the transverse relaxation rate proportional to the intracellular iron concentration, *[M-SPION]* is the concentration of intracellular iron on agarose gel and *r*_2_ is the transverse relaxivity of M-SPIONs.

The complement of the results was used to calculate the number of nanoparticles internalized into granulocytes using Equation (2):(2)$$Number\;of\;M-SPION=\frac{6\times iron\;load\times at\_m}{\pi \times \rho_{M-SPION}\times M_{Fe}\times \emptyset_{M-SPION}},$$
where iron load is the iron loaded into granulocytes by internalization (mass), at_m is the atomic mass, $\rho_{M-SPION}$ is the iron oxide density, $M_{Fe}$ is the molecular weight of iron and $\emptyset_{M-SPION}$ is the diameter of M-SPIONs.

#### 4.11.2. Quantification of Iron Load by ICP-MS

The quantification of the iron load of the granulocyte labeling with M-SPIONs by ICP-MS used 2 × 10^6^ cells/mL dispersed in 1 mL of DMEM and 1 mL of nitric acid (37%) for digestion over 4 h at 70 °C. The digested samples were diluted 10 times with Milli-Q^®^ water (EMD Millipore Corporation, Bedford MA, USA) and were analyzed with ICP-MS equipment (Perkin Elmer Nexion 350×, PerkinElmer Corporation, USA) to determine the iron content of each sample. Sample measurements were performed in quintuplicate, and quantification was based on a calibration curve using certified standard iron (NexION #N8145054) at the following concentrations: 5, 10, 20, 30 and 40 ppb.

#### 4.11.3. Quantification of Iron Load by NIRF

The quantification of iron load by NIRF was performed after granulocyte labeling with 2 × 10^6^ cells dispersed in 1 mL of DMEM placed in an Eppendorf tube. The NIRF signal of the triplicate samples was acquired by applying an excitation of 750 nm, registered in a range of emission of 810–875 nm, using an exposure time of 2.5 ms, binning of 4 and f/stop of 2. The sample images were acquired using IVIS^®^ Lumina LT Series III equipment (Xenogen Corp, PerkinElmer. CA, USA), and the fluorescence intensity signals were analyzed in absolute radiation units (photons/s). For the quantification analysis, the calibration curve was constructed using iron concentrations of 0.5, 1, 2, 8, 10 and 15 µg Fe/mL, and the fluorescence intensity signal of the samples was compared to a calibration curve to quantify the iron mass internalized by isolated granulocytes.

### 4.12. Statistics Analyses

The experimental data are presented as the mean and standard deviation. For comparison of the multiple isolation methods, differences between the means of the groups were tested by a nonparametric test for unpaired samples (Kruskal-Wallis test), following the post hoc analysis corrected by Bonferroni. The difference between two independent sample means was tested by a nonparametric Mann-Whitney test. Significant results were considered a *p*-value of less than 0.05. Statistical tests were performed using JASP v. 0.10.2.0 (JASP Team, 2019, Amsterdam, The Netherlands).

## Figures and Tables

**Figure 1 molecules-25-00765-f001:**
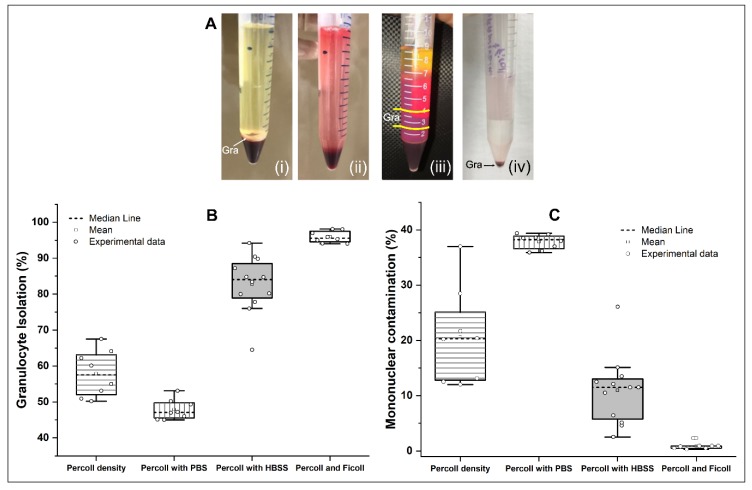
Purity and contaminant analysis of granulocyte isolation comparing the four methods of isolation. (**A**) Images at the end of each isolation method: (i) method I—Percoll with different gradients, (ii) method II—Percoll with phosphate buffer solution (PBS), (iii) method III—Percoll with Hank’s balanced salt solution (HBSS) and (iv) method IV—Percoll and Ficoll. (**B**) Boxplot of the purity of the granulocyte isolation between methods. (**C**) Boxplot of the contamination of granulocyte isolations between methods. Abbreviations Gra: granulocytes.

**Figure 2 molecules-25-00765-f002:**
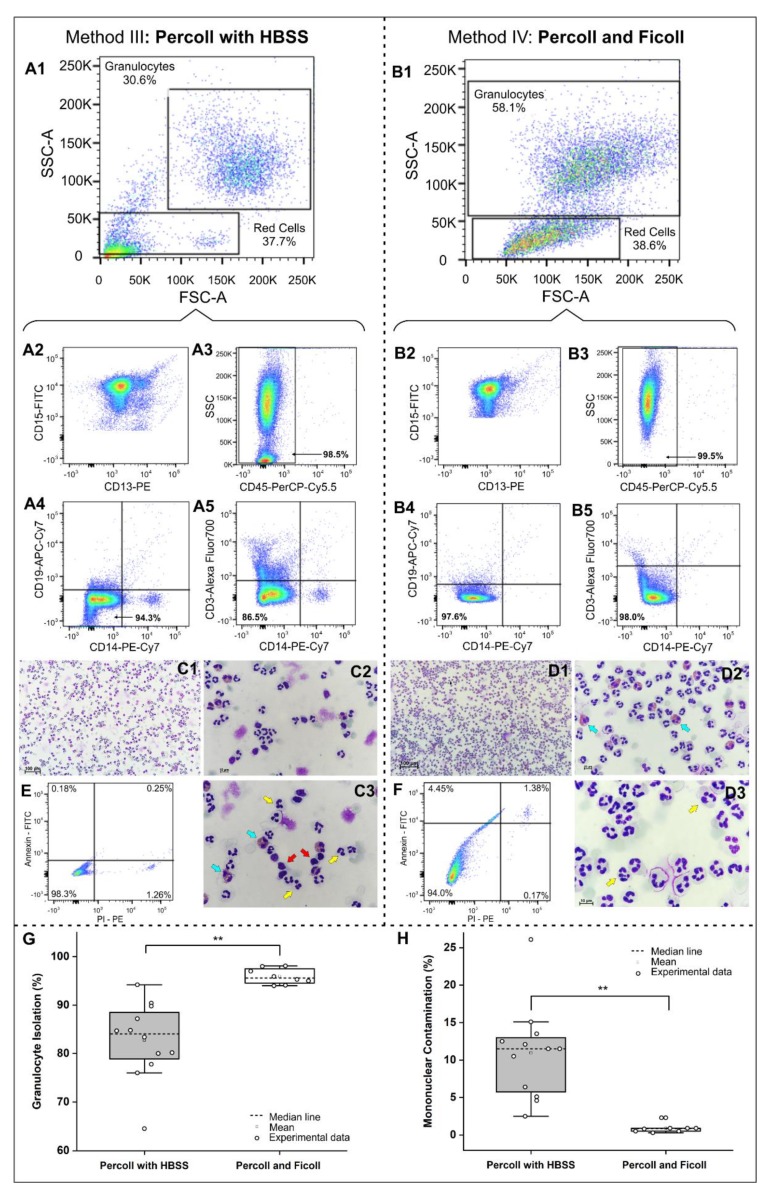
Analysis of the purity of isolated granulocytes and their morphological appearance (fast panoptic) between isolation methods III and IV. (**A1**,**B1**) Forward-scatter (FSC) vs. side-scatter (SSC) plots represent the isolated neutrophil population. Neutrophils isolated by methods III and IV were labeled with a mixture of antibodies and analyzed for the expression of (**A2**,**B2**) CD15 and CD13, (**A3**,**B3**) CD45 (as positive markers) and (**A4**,**B4**) CD19 and CD14 and (**A5**,**B5**) CD3 and CD14 (as negative markers). (**C1**–**3**,**D1**–**3**) Fast panoptic staining represents the morphological differences between neutrophils (indicated by the yellow arrows), contaminants (indicated by the red arrow) and the other granulocytes (indicated by blue arrows) under 10×, 20× and 40×. (**E**,**F**) Cell viability analysis by Annexin-FITC and propidium iodine-phycoerythrin (PI-PE). (**G**) Boxplot of the isolated granulocyte purity analysis for different methods. (**H**) Boxplot of the mononuclear contamination analysis for different methods. Representative pictures are shown from more than 10 slides analyzed.

**Figure 3 molecules-25-00765-f003:**
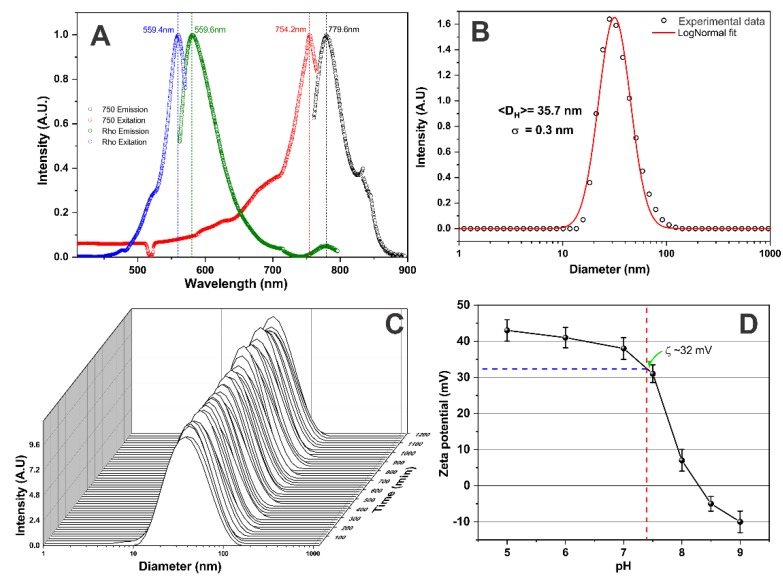
M-SPION characterizations: (**A**) M-SPION excitation/emission spectrum, showing the double fluorescence. (**B**) The curve of M-SPION size polydispersion. (**C**) The temporal stability analysis of M-SPIONs dispersed in culture medium (DMEM with 10% FBS). (**D**) Zeta potential as a function of the pH of the M-SPION solution.

**Figure 4 molecules-25-00765-f004:**
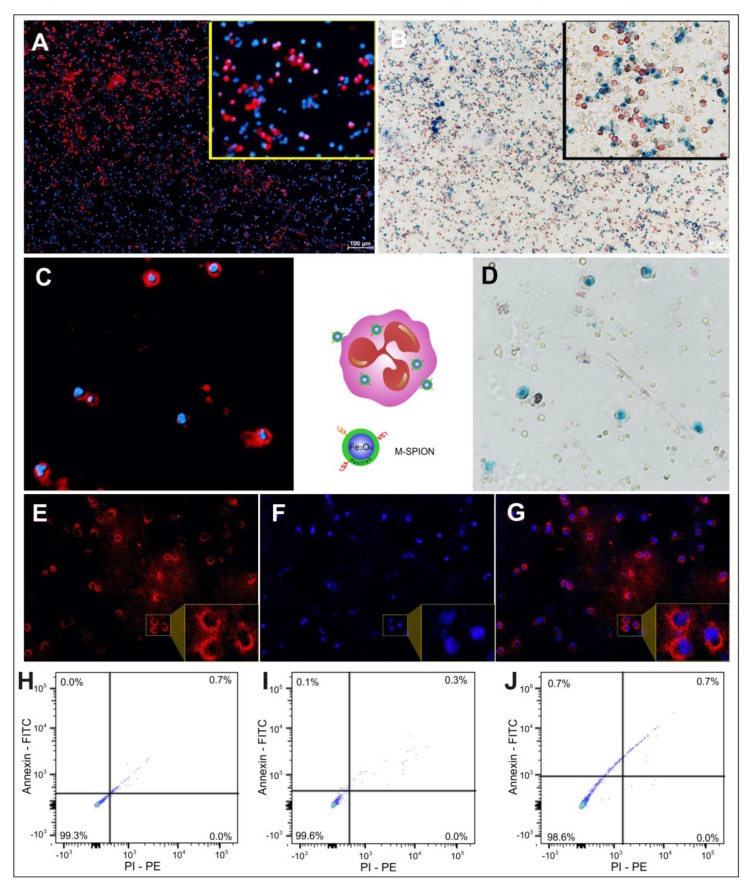
Fluorescence and brightfield microscopy and the viability of granulocytes labeled with M-SPIONs. (**A**) Fluorescence microscopy image of granulocytes labeled with M-SPIONs (4x). (**B**) Brightfield microscopy image of the granulocytes labeled with M-SPIONs and stained with Nuclear Fast Red and Prussian blue (4x) (**C**,**D**) Images corresponding to the same field of view of the granulocytes labeled with M-SPIONs by fluorescence and brightfield microscopy (4x). (**E**) Rhodamine imaged by fluorescence microscopy. (**F**) The corresponding image of granulocyte nuclear staining by DAPI. (**G**) Merged images of rhodamine/DAPI. The inset images (**A**,**B**,**E**–**G**) correspond to the amplified view of a selected area.(**H**–**J**) Viability was evaluated with M-SPION concentrations of 10, 30 and 50 μg Fe/mL respectively, with double negativity for Annexin-FITC and PI-PE.

**Figure 5 molecules-25-00765-f005:**
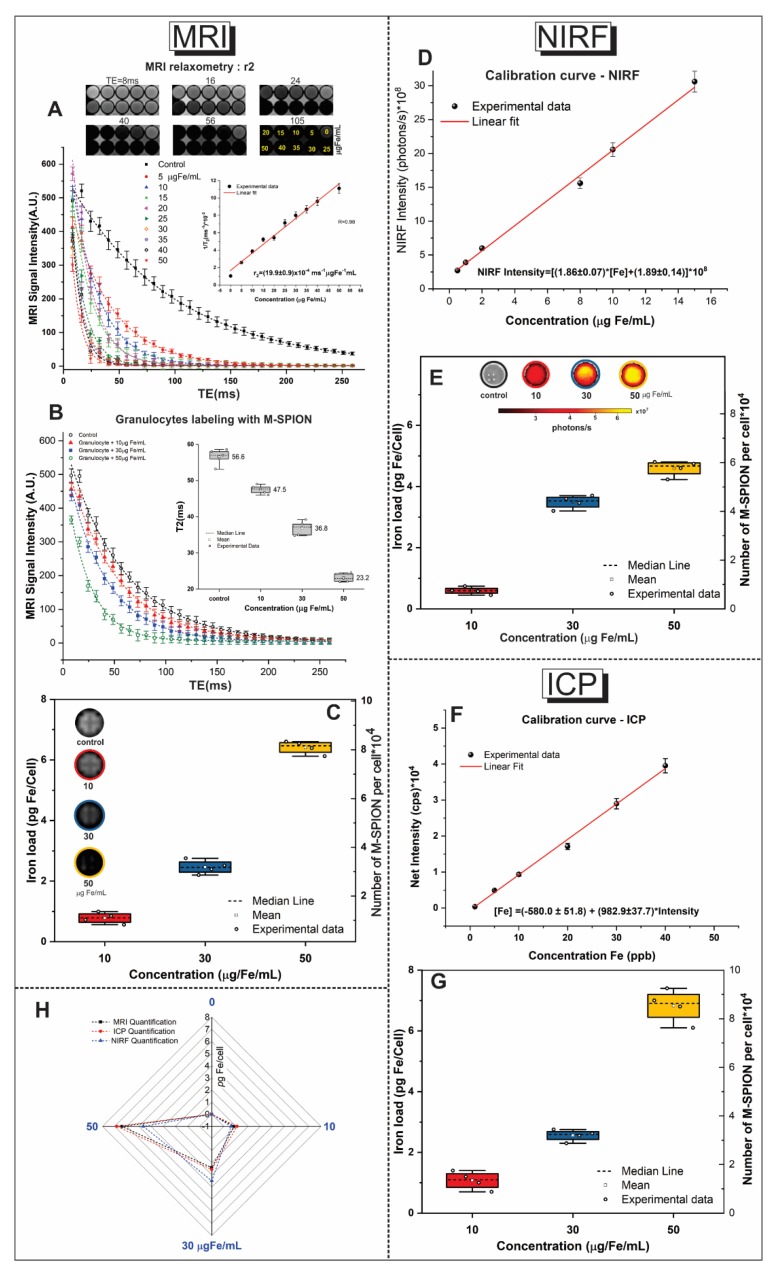
The quantification of M-SPIONs internalization into granulocytes. (**A**) MRI of the phantom containing different M-SPION concentrations and graph demonstrating the transverse relaxation curves of the signal intensity MRI as a function of echo time for M-SPION concentrations of 5, 10, 15, 20, 25, 30, 35, 40 and 50 µg Fe/mL. The inset image shows the r2 value determined from the linear adjustment of inverse transverse relaxation time as a function of iron concentration. (**B**) The graph of the signal intensity of MRI as a function of echo time for the samples of granulocytes labeled with M-SPION concentrations of 10, 30 and 50 µg Fe/mL. The inset shows the boxplot of the T2 value as a function of iron concentration used in each sample. (**C**) The boxplot of the iron load quantified by the MRI of the granulocytes labeled with M-SPION concentrations of 10, 30 and 50 µg Fe/mL in *p*g Fe per cell and number of M-SPIONs/cell. (**D**) The graph of the calibration curve of the signal intensity of near-infrared fluorescence (NIRF) as a function of iron concentration (µg Fe/mL), with the experimental data, the linear fit line and the equation used for the adjustment. (**E**) The boxplot of the iron load quantified by NIRF of the granulocytes labeled with M-SPION concentrations of 10, 30 and 50 µg Fe/mL in *p*g Fe/cell and the number of M-SPIONs/cell, calculated from the NIRF calibration curve. (**F**) The calibration curve of inductively coupled plasma mass spectrometry (ICP-MS), showing the net intensity (cps) as a function of iron concentration (ppb). The experimental data showed a linear fit. (**G**) The boxplot of the iron load quantified by ICP-MS of the granulocytes labeled with M-SPION concentrations of 10, 30 and 50 µg Fe/mL in *p*g Fe/cell and the number of M-SPIONs/cell. (**H**) Radar graph of the iron load quantified by MRI, NIRF and ICP-MS of granulocytes labeled with M-SPION concentrations of 10, 30 and 50 µg Fe/mL in *p*g Fe/cell.

**Figure 6 molecules-25-00765-f006:**
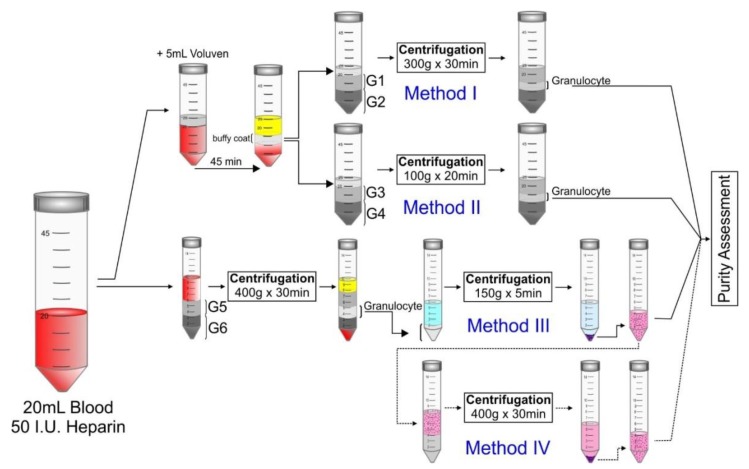
Schematic representation of the different granulocyte isolation methods (I-IV) tested on a blood sample for purity assessment. **Method I**—Granulocyte isolation with Percoll density gradients. **Method II**—Granulocyte isolation with Percoll density gradients and PBS. **Method III**—Granulocyte isolation with Percoll density gradients and Hanks′ solution. **Method IV**—Granulocyte isolation with Percoll gradients and Ficoll. Abbreviations: G1–G6—different gradient solutions of Percoll.

**Table 1 molecules-25-00765-t001:** Percentage mean purity of granulocyte isolation in each method.

Method	Mean	SD	N
Method I: Percoll gradients	57.89	6.477	8
Method II: Percoll with PBS	47.86	2.814	8
Method III: Percoll with HBSS	82.75	7.890	12
Method IV: Percoll and Ficoll	95.92	1.625	8

SD: standard deviation. PBS: phosphate buffer solution. HBSS: Hank’s balanced salt solution.

**Table 2 molecules-25-00765-t002:** Post hoc test—method comparisons.

Methods		Mean Difference	SE	t	p
Method I	Method II	10.03	2.867	3.496	0.008
	Method III	−24.86	2.617	−9.499	<0.001
	Method IV	−38.04	2.867	−13.266	<0.001
Method II	Method III	−34.89	2.617	−13.329	<0.001
	Method IV	−48.06	2.867	−16.762	<0.001
Method III	Method IV	−13.18	2.617	−5.033	<0.001

SE: standard error. t: student t-value. p: probability value.

**Table 3 molecules-25-00765-t003:** Percentage mean contamination (mononuclear) of each method.

Methods	Mean	SD	N
Method I: Percoll gradients	20.675	8.690	8
Method II: Percoll with PBS	37.850	1.335	8
Method III: Percoll with HBSS	10.950	6.212	12
Method IV: Percoll and Ficoll	0.887	0.615	8

SD: standard deviation.

**Table 4 molecules-25-00765-t004:** Post hoc test—method comparisons.

Methods		Mean Difference	SE	t	p
Method I	Method II	−17.175	2.750	−6.245	<0.001
	Method III	9.725	2.511	3.874	0.003
	Method IV	19.788	2.750	7.195	<0.001
Method II	Method III	26.900	2.511	10.715	<0.001
	Method IV	36.962	2.750	13.440	<0.001
Method III	Method IV	10.063	2.511	4.008	0.002

SE: standard error.

**Table 5 molecules-25-00765-t005:** Purity analysis of granulocytes isolated by methods III and IV.

Methods	N	Mean	SD	p
Method III: Percoll with HBSS	12	82.75	7.890	<0.001
Method IV: Percoll and Ficoll	8	95.92	1.625	

Mann–Whitney U test. SD: standard deviation.

**Table 6 molecules-25-00765-t006:** Mononuclear contamination analysis of granulocytes isolated by methods III and IV.

Methods	N	Mean	SD	p
Method III: Percoll with HBSS	12	10.950	6.212	<0.001
Method IV: Percoll and Ficoll	8	0.887	0.615	

Mann–Whitney U test. SD: standard deviation.

**Table 7 molecules-25-00765-t007:** The iron mass per cell and number of M-SPIONs internalized per granulocyte labeled with the given M-SPION concentrations in µg Fe/mL, determined using MRI, ICP-MS and NIRF.

[Fe]	MRI		ICP-MS		NIRF	
(µg/mL)	Mass (pg Fe/cell)	N of M-SPIONs *10^4^/cell	Mass (pg Fe/cell)	N of M-SPIONs *10^4^/cell	Mass (pg Fe/cell)	N of M-SPIONs *10^4^/cell
10	0.83 ± 0.11	1.04 ± 0.14	1.08 ± 0.26	1.35 ± 0.32	0.59 ± 0.10	0.74 ± 0.13
30	2.41 ± 0.15	3.03 ± 0.19	2.56 ± 0.16	3.21 ± 0.20	3.49 ± 0.19	4.39 ± 0.23
50	6.40 ± 0.18	8.06 ± 0.22	6.83 ± 0.47	8.56 ± 0.59	4.64 ± 0.14	5.84 ± 0.18

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
