# Peer review of "Methods of Granulocyte Isolation from Human Blood and Labeling with Multimodal Superparamagnetic Iron Oxide Nanoparticles"

_molecules, 2020, doi:10.3390/molecules25040765_

Round 1
Reviewer 1 Report
The paper shows the application of M-SPION for the detection of inflammation in granulocytes. Comprehensive results have been presented and appropriately described. The only weak point is the study of nanoparticles. Microscopic control of labeled with appropriate dyes nanoparticles does not always mean that nanoparticles have entered the cells. It is a pity that the authors did not examine their samples with M-SPION by other methods, for example electron spin resonance. Using this method it is possible to check the quality of such nanoparticles, their stability over a long period of time and to determine directly their concentration, also in cells.
Authors should include the characterization of the nanoparticles in the manuscript.
Author Response
Reviewer #1
The paper shows the application of M-SPION for the detection of inflammation in granulocytes. Comprehensive results have been presented and appropriately described.
The only weak point is the study of nanoparticles. Microscopic control of labeled with appropriate dyes nanoparticles does not always mean that nanoparticles have entered the cells. It is a pity that the authors did not examine their samples with M-SPION by other methods, for example electron spin resonance. Using this method it is possible to check the quality of such nanoparticles, their stability over a long period of time and to determine directly their concentration, also in cells.
Answer: Thank you for your observation. We understand that the ESR technique mentioned is useful for SPION evaluation, but this technique is not accessible in our lab, although we have performed quantification studies by this technique in a previous study [1]. The granulocyte labeling with M-SPION was evaluated by different techniques: brightfield microscopy and fluorescence imaging (qualitative evaluations) and ICP-MS, MRI, and NIRF (quantitative evaluations).
One of the qualitative evaluations of granulocyte labeling performed in our study was consistent with most previous studies that labeled granulocytes or other types of cells with SPIONs, namely, Prussian blue staining [2-8]. However, as the SPIONs used in this study had multimodal features, qualitative evaluation was also performed by fluorescence imaging (Figure 3E-G). In addition, corresponding images using both techniques, as shown in Figure 3C and D, enabled the use of the same visual field to verify the granulocytes labeled with SPIONs and observe that the SPIONS were mainly in the cytoplasm by fluorescence imaging. This technique was also performed in other studies that used similar SPIONs for cell labeling [9-12].
The granulocytes labeled with M-SPIONs were corroborated by quantitative evaluation using ICP-MS, MRI, and NIRF at the same M-SPION concentrations tested in this study (Figure 4). Through MRI, it was possible to observe T2*-weighted hypointense images as a function of the SPION concentration used in the labeling (Figure 4C), which corroborated the images visualized by the NIRF technique and the quantification of this technique (Figure 4E). Finally, gold standard quantification was also by ICP-MS (Figure 4G), in which the values found by the three techniques mentioned (Table 7) were correlated as a function of SPION concentration with regard to the sensitivity of each technique.
Regarding the SPION stability, we incorporated the suggestion to include this information, as this analysis was performed in our study as additional verification that the nanoparticles used did not change during the labeling process in the cell culture medium, although we did not previously include these results.
Therefore, the evaluation of M-SPION stability by dynamic light scattering (DLS) was added to the manuscript, considering the cell culture medium used for the labeling process (DMEM+10%FBS), as the medium containing the nanoparticles may interfere with the forces involved in their stability. The labeling of granulocytes was performed in a period of 4 hours, but the stability evaluation was performed in a longer period (20 hours, Figure 3), thus ensuring a longer evaluation of nanoparticle behavior in the same medium where cell labeling was performed. The results showed a consistent distribution of the hydrodynamic diameter of M-SPION without evidence of agglomeration during the analyzed period, indicating adequate stability.
References
Gamarra, L.F.; daCosta-Filho, A.J.; Mamani, J.B.; de Cassia Ruiz, R.; Pavon, L.F.; Sibov, T.T.; Vieira, E.D.; Silva, A.C.; Pontuschka, W.M.; Amaro, E., Jr. Ferromagnetic resonance for the quantification of superparamagnetic iron oxide nanoparticles in biological materials. International journal of nanomedicine 2010, 5, 203-211. Baraki, H.; Zinne, N.; Wedekind, D.; Meier, M.; Bleich, A.; Glage, S.; Hedrich, H.J.; Kutschka, I.; Haverich, A. Magnetic resonance imaging of soft tissue infection with iron oxide labeled granulocytes in a rat model. PLoS One 2012, 7, e51770. Han, S.; Xing, W.; Lu, H.; Han, H. Early diagnostic method for sepsis based on neutrophil MR imaging. Radiology of Infectious Diseases 2015, 2, 33-39, Shanhua, H.; Huijing, H.; Moon, M.J.; Heo, S.H.; Lim, H.S.; Park, I.K.; Cho, C.S.; Kwak, S.H.; Jeong, Y.Y. MR detection of LPS-induced neutrophil activation using mannan-coated superparamagnetic iron oxide nanoparticles. Mol Imaging Biol 2013, 15, 685-692 Naseroleslami, M.; Aboutaleb, N.; Parivar, K. The effects of superparamagnetic iron oxide nanoparticles-labeled mesenchymal stem cells in the presence of a magnetic field on attenuation of injury after heart failure. Drug delivery and translational research 2018, 8, 1214-1225, Parsa, H.; Shamsasenjan, K.; Movassaghpour, A.; Akbarzadeh, P.; Amoghli Tabrizi, B.; Dehdilani, N.; Lotfinegad, P.; Soleimanloo, F. Effect of Superparamagnetic Iron Oxide Nanoparticles-Labeling on Mouse Embryonic Stem Cells. Cell J 2015, 17, 221-230, Zare, S.; Mehrabani, D.; Jalli, R.; Saeedi Moghadam, M.; Manafi, N.; Mehrabani, G.; Jamhiri, I.; Ahadian, S. MRI-Tracking of Dental Pulp Stem Cells In Vitro and In Vivo Using Dextran-Coated Superparamagnetic Iron Oxide Nanoparticles. J Clin Med 2019, 8, 1418 Zou, Q.; Zhang, C.J.; Yan, Y.Z.; Min, Z.J.; Li, C.S. MUC-1 aptamer targeted superparamagnetic iron oxide nanoparticles for magnetic resonance imaging of pancreatic cancer in vivo and in vitro experiment. Journal of cellular biochemistry 2019, 120, 18650-18658. da Silva, H.R.; Mamani, J.B.; Nucci, M.P.; Nucci, L.P.; Kondo, A.T.; Fantacini, D.M.C.; de Souza, L.E.B.; Picanço-Castro, V.; Covas, D.T.; Kutner, J.M., et al. Triple-modal imaging of stem-cells labeled with multimodal nanoparticles, applied in a stroke model. World journal of stem cells 2019, 11, 100-123.
10.Li, W.-J.; Wang, Y.; Liu, Y.; Wu, T.; Cai, W.-L.; Shuai, X.-T.; Hong, G.-B. Preliminary Study of MR and Fluorescence Dual-mode Imaging: Combined Macrophage-Targeted and Superparamagnetic Polymeric Micelles. Int J Med Sci 2018, 15, 129-141
11.Pala, A.; Liberatore, M.; D'Elia, P.; Nepi, F.; Megna, V.; Mastantuono, M.; Al-Nahhas, A.; Rubello, D.; Barteri, M. Labelling of granulocytes by phagocytic engulfment with 64Cu-labelled chitosan-coated magnetic nanoparticles. Mol Imaging Biol 2012, 14, 593-598,
12.Alvarim, L.T.; Nucci, L.P.; Mamani, J.B.; Marti, L.C.; Aguiar, M.F.; Silva, H.R.; Silva, G.S.; Nucci-da-Silva, M.P.; DelBel, E.A.; Gamarra, L.F. Therapeutics with SPION-labeled stem cells for the main diseases related to brain aging: a systematic review. International journal of nanomedicine 2014, 9, 3749-3770,
Authors should include the characterization of the nanoparticles in the manuscript.
Answer: Thank you for this suggestion. We added information to the manuscript (items 2.3, 3 and 4.7) on the optical characterization, the SPION hydrodynamic size polydispersion analysis, the temporal stability of the SPIONs, and the zeta potential measurements. The first version of the submitted manuscript did included characterization of the M-SPION relaxometry by MRI. Thus, the M-SPION characterizations added to the manuscript in response to this valuable suggestion are enough to support the proposed objective in this study.

Reviewer 2 Report
In this paper, Alvieri and collaborators tested four different methods for granulocyte isolation and subsequent labelling with SPION, showing that Percoll-Ficoll method was the one that achieved higher purity and viability. Then, they demonstrated that labeling granulocytes with these M-SPION provides high internalized iron load and low toxicity by MRI, NIRF and ICP techniques.
The study is globally well organized and clear. Figures nicely present the results.
Nevertheless, I have some concerns:
Last sentence abstract seems overstatement since the authors have not performed in vivo molecular MRI using their probe. More generally, the clinical relevance of this work remains unclear: the sensitivity of MRI is probably too low at clinically relevant field strength and resolution to detect autologous granulocytes even if the authors state that is detected by MRI and NIRF. This should be further discussed by the authors.
The added value of granulocytes labeling compared to direct targeting of SPION (such as MPIO) should be discussed, including about the plausibility of clinical translation.
It would have been relevant to compare the labeling efficiency of different type of SPIO and under different experimental conditions (inflammatory conditions?).
What is the fate of the SPION after internalization?
In vivo experiments are lacking: is it possible to detect labeled granulocytes in an experimental model of inflammation (such as intracerebral LPS or TNF injection)?
Minor:
-Precise the multimodal capacity of the agent used in the abstract: Fluorescence + magnetic since this information is only present in the full text.
-Improve contrast quality of figure 3 panel E-F-G
-Attention, since the centrigugation rates seem differ between the method text and figure 5: Notable on the two first granulocyte isolation methods : method 1 (percoll) and 2 (PBS). Please clarify with are the correct ones (either 150g and 300g of the method section or 300g and 100g shown in the figure).
-Please modify RPM speed to international unit (g or RCF) on method 4.3
-Indicate final reaction volume used (100 microliters?) for flow cytometry analysis in method 4.4 since you only precise the amount of microlitres used for antibody labelling.
-Finally, correct some English- editing- mistakes:
Quite few sentences in the introduction are misleading and the final message is not clear. For instance, one example: “The clinical diagnosis consists of biochemical and radiological examination, but these methods can produce false negatives or but these methods can fail.”
Figure legend 3 Misspelling in Prussian blue
FBS is sometimes written not abbreviated or like SFB
FBS in method 4.7
Prussian blue in method 4.8
Correct and clarify the English language in the author contributors section like: Investigation, Data curation, Visualization, Draft preparation and writing
Abbreviations section: Sodium chloride, Millimeter
Author Response
Reviewer #2
In this paper, Alvieri and collaborators tested four different methods for granulocyte isolation and subsequent labelling with SPION, showing that Percoll-Ficoll method was the one that achieved higher purity and viability. Then, they demonstrated that labeling granulocytes with these M-SPION provides high internalized iron load and low toxicity by MRI, NIRF and ICP techniques.
The study is globally well organized and clear. Figures nicely present the results
Nevertheless, I have some concerns:
Last sentence abstract seems overstatement since the authors have not performed in vivo molecular MRI using their probe. More generally, the clinical relevance of this work remains unclear: the sensitivity of MRI is probably too low at clinically relevant field strength and resolution to detect autologous granulocytes even if the authors state that is detected by MRI and NIRF. This should be further discussed by the authors.
Answer: Thank you for your consideration. Regarding the final sentence of the abstract, we agree that it was misplaced, implying that the present study also included in vivo experiments, but in fact, the study focuses on methodological evidence from in vitro experiments for future in vivo applications. We also understand the limitations of the use of immune cells exogenously labeled with nanoparticles in the detection of inflammation processes. Given these limitations, this study aimed to improve some important aspects to increase the sensitivity of detection by MRI, such as improving the isolation and purification techniques to increase both the yield and the purity of isolated cells and developing strategies for immune cell labeling to increase the internalized SPION load without affecting cell viability. The study results showed a higher yield and a higher degree of purity after cell isolation as well as a higher internalized SPION load compared to the literature [1,2], showing that 500 times more SPIONS were internalized without affecting cell viability. In addition to improving MRI sensitivity in our study, the M-SPIONs fluoresce in the infrared spectrum, enabling optical image detection.
Some relevant factors for improving the sensitivity of MRI detection have already been discussed in the manuscript. However, we add this information with greater emphasis on in vivo detection to emphasize the viability of using this technique in an in vivo model, as has been shown by other studies that reported lower in vitro detection characteristics than those found in our study but nevertheless achieved in vivo application and detection [3-6].
In our future studies, these cells exogenously labeled with multimodal nanoparticles will be applied in inflammation models, making it possible to verify the aspects questioned in the review, such as detection sensitivity in an in vivo model and SPION elimination processes.
The added value of granulocytes labeling compared to direct targeting of SPION (such as MPIO) should be discussed, including about the plausibility of clinical translation.
Answer: There are different strategies for detecting inflammation processes. Our study presented one in which immune cells are isolated, exogenously labeled with nanoparticles, and systemically injected (autologous cells) for homing and monitoring by noninvasive imaging techniques [2,6-9]. Another strategy is to perform endogenous labeling of monocyte/macrophage strains using nanoparticles and MPIO [7,10,11]. Each technique has its advantages and disadvantages. The homing of exogenously labeled immunological cells has shown high specificity because SPION-labeled immune cells are less susceptible to nonspecific targeting and allow direct imaging of inflammatory immune process interactions considering all processes involved in the regulation of endothelial and blood binding to autologous cell diapedesis[7]. Some limiting factors of this technique are related to the detection sensitivity of the MRI technique but can be improved by (i) increasing the efficiency of isolation purification, that is, increasing the yield and purity of the isolated cells, and (ii) using strategies for immunological cell labeling to increase the internalized SPION load without affecting cell viability. These aspects were addressed in this study, and the results showed a higher degree of purity and higher yield of isolated cells than those obtained in the literature [6,9,12-17], as well as greater SPION load internalization compared to that in the literature [1-3]. These aspects both improve MRI sensitivity, but another important feature of the SPIONs used in this study is their multimodal character, enabling detection by near-infrared fluorescence (NIRF) in vivo and by optical fluorescence techniques in vitro.
Regarding endogenous labeling techniques using ultrasmall superparamagnetic iron oxides (USPIO), studies have shown nonspecificity due to their absorption by monocytes, macrophages and liver Kupffer cells and their accumulation in lymphoid organs such as the lymph nodes, liver and spleen in addition to the injury site (target) [7,10,11]. The detection of endogenous USPIO by MRI is mainly characterized by increased vascular permeability and site hyperemia due to the intravenous presence of USPIO and recruitment of the number of macrophages that have internalized endogenous USPIO [7,8,10,11,18-21], enabling the detection of acute and chronic inflammation [8].
The use of MPIO as an endogenous labeling technique is characterized by the fact that they are specific microparticles for vascular adhesion molecules that present as inflammatory markers, with high MRI contrast, absence of extravasation and rapid blood clearance [22,23]; however, they have a disadvantage in MRI, namely, hemorrhages and blood oxygen level dependent (BOLD) effects induce signal voids on T2*-weighted images that are comparable to MPIO-induced signal voids and could therefore lead to false positive results [7]. Other limitations and disadvantages to the application of MPIO have also been highlighted, such as nonbiodegradable coating, iron toxicity potential, and accumulation in the reticulo-endothelial system; MPIO targets are limited to proteins expressed by the endothelium, such as cell adhesion molecules able to enter, for example, the brain parenchyma, and a high dose of iron is necessary in experimental studies [7]. Another study points out that MPIOs have limitations that make them inadequate for clinical application such as rapid sedimentation, slow degradation and mechanical retention in the organs. [10,24,25]. For some applications, the size of these particles would prevent delivery to the place of interest. [10]. A discussion of these points was added to the manuscript.
It would have been relevant to compare the labeling efficiency of different type of SPIO and under different experimental conditions (inflammatory conditions?).
Answer: Thank you for your suggestion. However, the focus of this in vitro study was to find the best (purity and yield) granulocyte isolation technique, with the aim of labeling with multimodal nanoparticles detectable by MRI-NIRF imaging, qualitative and quantitative evaluation of their internalized M-SPION loads, to prepare a promising candidate for use in future studies on inflammation/infection process detection by optical and MRI techniques.
Regarding SPION comparison, other SPIONs were used in a previous study[26-35], and the M-SPIONs showed better characteristics for this kind of application: adequate stability, no toxicity, and trimodal properties including fluorescence for detection by MRI, NIRF, and the visible light spectrum. The best concentrations for in vitro application were analyzed. In addition, future in vivo studies will analyze models of inflammatory processes.
What is the fate of the SPION after internalization?
Answer: This study performed only in vitro experiments, but there are a growing number of studies dedicated to understanding SPION removal after labeling of cells of interest and subsequent administration in an in vivo model [36-40]. SPION elimination pathways involve various different organs depending on the SPION physicochemical characteristics, such as size, shape, coating, and surface charge, as well as experimental conditions [41], but the combined results of these studies reveal that the process of elimination occurs mainly through the liver and spleen after accumulation [20,36-43].
A future step in this work will be the application of these M-SPION-labeled immune cells in preclinical models of inflammatory processes, which will enable the effective evaluation of SPION elimination after administration. This question raised by the reviewer is of great relevance, especially in clinical application.
In vivo experiments are lacking: is it possible to detect labeled granulocytes in an experimental model of inflammation (such as intracerebral LPS or TNF injection)?
Answer: Thank you for your observation. We modified a few phrases of the manuscript to clarify that this study performed only in vitro experiments aimed to find the best (purity and yield) granulocyte isolation technique, with the aim of labeling with multimodal nanoparticles detectable by MRI-NIRF imaging and qualitative and quantitative evaluation of their internalized M-SPION loads. Therefore, after achieving this goal, the same M-SPIONs could be considered a promising candidate for future research on inflammation/infection process detection by optical and MRI techniques, as the reviewer comments. Other studies have reported SPION labeling in granulocyte cells for inflammation process detection[3,44].
Precise the multimodal capacity of the agent used in the abstract: Fluorescence + magnetic since this information is only present in the full text.
Answer: Thank you for the observation. We added information on the magnetic and fluorescent properties of M-SPIONs to the abstract.
Improve contrast quality of figure 3 panel E-F-G
Answer: Thank you for the suggestion. We improved the contrast quality of Figure 3.
Attention, since the centrifugation rates seem differ between the method text and figure 5: Notable on the two first granulocyte isolation methods : method 1 (percoll) and 2 (PBS). Please clarify with are the correct ones (either 150g and 300g of the method section or 300g and 100g shown in the figure)
Answer: Thank you for the observation. We corrected the typographical errors in the manuscript to match the values shown in Figure 6.
Please modify RPM speed to international unit (g or RCF) on method 4.3
Answer: Thank you for the suggestion. We modified the centrifugation speeds to international units in item 4.3 of the Methods section.
Indicate final reaction volume used (100 microliters?) for flow cytometry analysis in method 4.4 since you only precise the amount of microlitres used for antibody labelling.
Answer: Thank you for the suggestion. The final reaction volume used was added to the manuscript (Item 4.4 of the Methods section).
Finally, correct some English- editing- mistakes:
Quite few sentences in the introduction are misleading and the final message is not clear. For instance, one example: “The clinical diagnosis consists of biochemical and radiological examination, but these methods can produce false negatives or but these methods can fail.”
-Figure legend 3 Misspelling in Prussian blue
-FBS is sometimes written not abbreviated or like SFB
-FBS in method 4.7
-Prussian blue in method 4.8
Answer: Thank you for the suggestions. The sentence mentioned by the reviewer in the introduction section of the manuscript was corrected, as well as all the mistakes indicated.
Correct and clarify the English language in the author contributors section like: Investigation, Data curation, Visualization, Draft preparation and writing
Answer: Thank you for the suggestion. We modified these items in the author contribution section.
Abbreviations section: Sodium chloride, Millimeter
Answer: Thank you for the suggestion. We modified these items in the abbreviations section.
Reference:
1. Han, S.; Xing, W.; Lu, H.; Han, H. Early diagnostic method for sepsis based on neutrophil MR imaging. Radiology of Infectious Diseases 2015, 2, 33-39, doi:https://doi.org/10.1016/j.jrid.2015.05.002. Shanhua, H.; Huijing, H.; Moon, M.J.; Heo, S.H.; Lim, H.S.; Park, I.K.; Cho, C.S.; Kwak, S.H.; Jeong, Y.Y. MR detection of LPS-induced neutrophil activation using mannan-coated superparamagnetic iron oxide nanoparticles. Mol Imaging Biol 2013, 15, 685-692, doi:10.1007/s11307-013-0643-x. Krieg, F.M.; Andres, R.Y.; Winterhalter, K.H. Superparamagnetically labelled neutrophils as potential abscess-specific contrast agent for MRI. Magnetic Resonance Imaging 1995, 13, 393-400, doi:https://doi.org/10.1016/0730-725X(94)00111-F. Zelivyanskaya, M.L.; Nelson, J.A.; Poluektova, L.; Uberti, M.; Mellon, M.; Gendelman, H.E.; Boskal, M.D. Tracking superparamagnetic iron oxide labeled monocytes in brain by high-field magnetic resonance imaging. Journal of Neuroscience Research 2003, 73, 284-295, doi:10.1002/jnr.10693. Engberink, R.D.O.; van der Pol, S.M.A.; Walczak, P.; van der Toorn, A.; Viergever, M.A.; Dijkstra, C.D.; Bulte, J.W.M.; de Vries, H.E.; Blezer, E.L.A. Magnetic Resonance Imaging of Monocytes Labeled with Ultrasmall Superparamagnetic Particles of Iron Oxide Using Magnetoelectroporation in an Animal Model of Multiple Sclerosis. Molecular Imaging 2010, 9, 7290.2010.00016, doi:10.2310/7290.2010.00016. Baraki, H.; Zinne, N.; Wedekind, D.; Meier, M.; Bleich, A.; Glage, S.; Hedrich, H.J.; Kutschka, I.; Haverich, A. Magnetic resonance imaging of soft tissue infection with iron oxide labeled granulocytes in a rat model. PLoS One 2012, 7, e51770, doi:10.1371/journal.pone.0051770. Gauberti, M.; Montagne, A.; Quenault, A.; Vivien, D. Molecular magnetic resonance imaging of brain-immune interactions. Front Cell Neurosci 2014, 8, 389, doi:10.3389/fncel.2014.00389. Neuwelt, A.; Sidhu, N.; Hu, C.A.; Mlady, G.; Eberhardt, S.C.; Sillerud, L.O. Iron-based superparamagnetic nanoparticle contrast agents for MRI of infection and inflammation. AJR Am J Roentgenol 2015, 204, W302-313, doi:10.2214/ajr.14.12733. Pala, A.; Liberatore, M.; D'Elia, P.; Nepi, F.; Megna, V.; Mastantuono, M.; Al-Nahhas, A.; Rubello, D.; Barteri, M. Labelling of granulocytes by phagocytic engulfment with 64Cu-labelled chitosan-coated magnetic nanoparticles. Mol Imaging Biol 2012, 14, 593-598, doi:10.1007/s11307-011-0526-y. McAteer, M.A.; Schneider, J.E.; Ali, Z.A.; Warrick, N.; Bursill, C.A.; von zur Muhlen, C.; Greaves, D.R.; Neubauer, S.; Channon, K.M.; Choudhury, R.P. Magnetic resonance imaging of endothelial adhesion molecules in mouse atherosclerosis using dual-targeted microparticles of iron oxide. Arterioscler Thromb Vasc Biol 2008, 28, 77-83, doi:10.1161/atvbaha.107.145466. McAteer, M.A.; Sibson, N.R.; von Zur Muhlen, C.; Schneider, J.E.; Lowe, A.S.; Warrick, N.; Channon, K.M.; Anthony, D.C.; Choudhury, R.P. In vivo magnetic resonance imaging of acute brain inflammation using microparticles of iron oxide. Nat Med 2007, 13, 1253-1258, doi:10.1038/nm1631. Sursal, N.; Cakmak, A.; Yildiz, K. Neutrophil isolation from feline blood using discontinuous Percoll dilutions. Tierarztliche Praxis. Ausgabe K, Kleintiere/Heimtiere 2018, 46, 399-402, doi:10.1055/s-0038-1677404. Soltys, J.; Swain, S.D.; Sipes, K.M.; Nelson, L.K.; Hanson, A.J.; Kantele, J.M.; Jutila, M.A.; Quinn, M.T. Isolation of bovine neutrophils with biomagnetic beads: comparison with standard Percoll density gradient isolation methods. J Immunol Methods 1999, 226, 71-84. Kuhns, D.B.; Long Priel, D.A.; Chu, J.; Zarember, K.A. Isolation and Functional Analysis of Human Neutrophils. Curr Protoc Immunol 2015, 111, 7.23.21-16, doi:10.1002/0471142735.im0723s111. Hirz, T.; Dumontet, C. Neutrophil Isolation and Analysis to Determine their Role in Lymphoma Cell Sensitivity to Therapeutic Agents. J Vis Exp 2016, 10.3791/53846, e53846, doi:10.3791/53846. Zhou, L.; Somasundaram, R.; Nederhof, R.F.; Dijkstra, G.; Faber, K.N.; Peppelenbosch, M.P.; Fuhler, G.M. Impact of human granulocyte and monocyte isolation procedures on functional studies. Clin Vaccine Immunol 2012, 19, 1065-1074, doi:10.1128/cvi.05715-11. Pleskova, S.N.; Mikheeva, E.R.; Gornostaeva, E.E. The interaction between human blood neutrophil granulocytes and quantum dots. Micron 2018, 105, 82-92, doi:10.1016/j.micron.2017.11.011. Vermeij, E.A.; Koenders, M.I.; Bennink, M.B.; Crowe, L.A.; Maurizi, L.; Vallee, J.P.; Hofmann, H.; van den Berg, W.B.; van Lent, P.L.; van de Loo, F.A. The in-vivo use of superparamagnetic iron oxide nanoparticles to detect inflammation elicits a cytokine response but does not aggravate experimental arthritis. PLoS One 2015, 10, e0126687, doi:10.1371/journal.pone.0126687. Seyfer, P.; Pagenstecher, A.; Mandic, R.; Klose, K.J.; Heverhagen, J.T. Cancer and inflammation: differentiation by USPIO-enhanced MR imaging. Journal of magnetic resonance imaging : JMRI 2014, 39, 665-672, doi:10.1002/jmri.24200. Ashraf, S.; Taylor, A.; Sharkey, J.; Barrow, M.; Murray, P.; Wilm, B.; Poptani, H.; Rosseinsky, M.J.; Adams, D.J.; Lévy, R. In vivo fate of free and encapsulated iron oxide nanoparticles after injection of labelled stem cells. Nanoscale Advances 2019, 1, 367-377, doi:10.1039/C8NA00098K. Lefevre, S.; Ruimy, D.; Jehl, F.; Neuville, A.; Robert, P.; Sordet, C.; Ehlinger, M.; Dietemann, J.L.; Bierry, G. Septic arthritis: monitoring with USPIO-enhanced macrophage MR imaging. Radiology 2011, 258, 722-728, doi:10.1148/radiol.10101272. McAteer, M.A.; Sibson, N.R.; von Zur Muhlen, C.; Schneider, J.E.; Lowe, A.S.; Warrick, N.; Channon, K.M.; Anthony, D.C.; Choudhury, R.P. In vivo magnetic resonance imaging of acute brain inflammation using microparticles of iron oxide. Nature medicine 2007, 13, 1253-1258, doi:10.1038/nm1631. Shapiro, E.M.; Skrtic, S.; Sharer, K.; Hill, J.M.; Dunbar, C.E.; Koretsky, A.P. MRI detection of single particles for cellular imaging. Proceedings of the National Academy of Sciences of the United States of America 2004, 101, 10901-10906, doi:10.1073/pnas.0403918101. Perez-Balderas, F.; van Kasteren, S.I.; Aljabali, A.A.; Wals, K.; Serres, S.; Jefferson, A.; Sarmiento Soto, M.; Khrapitchev, A.A.; Larkin, J.R.; Bristow, C., et al. Covalent assembly of nanoparticles as a peptidase-degradable platform for molecular MRI. Nature communications 2017, 8, 14254, doi:10.1038/ncomms14254. Ilium, L.; Davis, S.S.; Wilson, C.G.; Thomas, N.W.; Frier, M.; Hardy, J.G. Blood clearance and organ deposition of intravenously administered colloidal particles. The effects of particle size, nature and shape. International Journal of Pharmaceutics 1982, 12, 135-146, doi:https://doi.org/10.1016/0378-5173(82)90113-2. Gamarra, L.F.; Pavon, L.F.; Marti, L.C.; Pontuschka, W.M.; Mamani, J.B.; Carneiro, S.M.; Camargo-Mathias, M.I.; Moreira-Filho, C.A.; Amaro, E. In vitro study of CD133 human stem cells labeled with superparamagnetic iron oxide nanoparticles. Nanomedicine: Nanotechnology, Biology and Medicine 2008, 4, 330-339, doi:https://doi.org/10.1016/j.nano.2008.05.002. Gamarra, L.F.; Pontuschka, W.M.; Amaro, E.; Costa-Filho, A.J.; Brito, G.E.S.; Vieira, E.D.; Carneiro, S.M.; Escriba, D.M.; Falleiros, A.M.F.; Salvador, V.L. Kinetics of elimination and distribution in blood and liver of biocompatible ferrofluids based on Fe3O4 nanoparticles: An EPR and XRF study. Materials Science and Engineering: C 2008, 28, 519-525, doi:https://doi.org/10.1016/j.msec.2007.06.005. Gamarra, L.F.; Pontuschka, W.M.; Mamani, J.B.; Cornejo, D.R.; Oliveira, T.R.; Vieira, E.D.; Costa-Filho, A.J.; Amaro Jr, E. Magnetic characterization by SQUID and FMR of a biocompatible ferrofluid based on Fe3O4. Journal of Physics: Condensed Matter 2009, 21, 115104, doi:10.1088/0953-8984/21/11/115104. Ferreira, R.V.; Pereira, I.L.S.; Cavalcante, L.C.D.; Gamarra, L.F.; Carneiro, S.M.; Amaro, E.; Fabris, J.D.; Domingues, R.Z.; Andrade, A.L. Synthesis and characterization of silica-coated nanoparticles of magnetite. Hyperfine Interactions 2010, 195, 265-274, doi:10.1007/s10751-009-0128-0. Gamarra, L.F.; Amaro, E.; Alves, S.; Soga, D.; Pontuschka, W.M.; Mamani, J.B.; Carneiro, S.M.; Brito, G.E.S.; Figueiredo Neto, A.M. Characterization of the Biocompatible Magnetic Colloid on the Basis of Fe3O4 Nanoparticles Coated with Dextran, Used as Contrast Agent in Magnetic Resonance Imaging. Journal of nanoscience and nanotechnology 2010, 10, 4145-4153, doi:10.1166/jnn.2010.2200. Gamarra, L.F.; Mamani, J.B.; Carneiro, S.M.; Fabris, J.D.; Ferreira, R.V.; Domingues, R.Z.; Cornejo, D.R.; Pontuschka, W.M.; Amaro, E. Characterization of Superparamagnetic Iron Oxide Coated with Silicone Used as Contrast Agent for Magnetic Resonance Image for the Gastrointestinal Tract. Journal of nanoscience and nanotechnology 2010, 10, 1153-1158, doi:10.1166/jnn.2010.1843. Sibov, T.T.; Miyaki, L.A.M.; Mamani, J.B.; Marti, L.C.; Sardinha, L.R.; Pavon, L.F.; Oliveira, D.M.d.; Cardenas, W.H.; Gamarra, L.F. Evaluation of umbilical cord mesenchymal stem cell labeling with superparamagnetic iron oxide nanoparticles coated with dextran and complexed with Poly-L-lysine. Einstein (Sao Paulo, Brazil) 2012, 10, 180-188. Silva, A.C.d.; Cabral, F.R.; Mamani, J.B.; Malheiros, J.M.; Polli, R.S.; Tannus, A.; Vidoto, E.; Martins, M.J.; Sibov, T.T.; Pavon, L.F., et al. Tumor growth analysis by magnetic resonance imaging of the C6 glioblastoma model with prospects for the assessment of magnetohyperthermia therapy. Einstein (Sao Paulo, Brazil) 2012, 10, 11-15. Mamani, J.B.; Costa-Filho, A.J.; Cornejo, D.R.; Vieira, E.D.; Gamarra, L.F. Synthesis and characterization of magnetite nanoparticles coated with lauric acid. Materials Characterization 2013, 81, 28-36, doi:https://doi.org/10.1016/j.matchar.2013.04.001. Mamani, J.B.; Gamarra, L.F.; Brito, G.E.d.S. Synthesis and characterization of Fe3O4 nanoparticles with perspectives in biomedical applications. Materials Research 2014, 17, 542-549. Kolosnjaj-Tabi, J.; Javed, Y.; Lartigue, L.; Volatron, J.; Elgrabli, D.; Marangon, I.; Pugliese, G.; Caron, B.; Figuerola, A.; Luciani, N., et al. The One Year Fate of Iron Oxide Coated Gold Nanoparticles in Mice. ACS Nano 2015, 9, 7925-7939, doi:10.1021/acsnano.5b00042. Feliu, N.; Docter, D.; Heine, M.; del Pino, P.; Ashraf, S.; Kolosnjaj-Tabi, J.; Macchiarini, P.; Nielsen, P.; Alloyeau, D.; Gazeau, F., et al. In vivo degeneration and the fate of inorganic nanoparticles. Chemical Society Reviews 2016, 45, 2440-2457, doi:10.1039/C5CS00699F. Kolosnjaj Tabi, J.; Volatron, J.; Gazeau, F. Basic Principles of In Vivo Distribution, Toxicity, and Degradation of Prospective Inorganic Nanoparticles for Imaging. 2017; 10.1007/978-3-319-42169-8_2pp. 9-41. Levy, M.; Luciani, N.; Alloyeau, D.; Elgrabli, D.; Deveaux, V.; Pechoux, C.; Chat, S.; Wang, G.; Vats, N.; Gendron, F., et al. Long term in vivo biotransformation of iron oxide nanoparticles. Biomaterials 2011, 32, 3988-3999, doi:10.1016/j.biomaterials.2011.02.031. Kolosnjaj-Tabi, J.; Lartigue, L.; Javed, Y.; Luciani, N.; Pellegrino, T.; Wilhelm, C.; Alloyeau, D.; Gazeau, F. Biotransformations of magnetic nanoparticles in the body. Nano Today 2016, 11, 280-284, doi:https://doi.org/10.1016/j.nantod.2015.10.001. Feng, Q.; Liu, Y.; Huang, J.; Chen, K.; Huang, J.; Xiao, K. Uptake, distribution, clearance, and toxicity of iron oxide nanoparticles with different sizes and coatings. Scientific reports 2018, 8, 2082, doi:10.1038/s41598-018-19628-z. Lee, M.J.; Veiseh, O.; Bhattarai, N.; Sun, C.; Hansen, S.J.; Ditzler, S.; Knoblaugh, S.; Lee, D.; Ellenbogen, R.; Zhang, M., et al. Rapid pharmacokinetic and biodistribution studies using cholorotoxin-conjugated iron oxide nanoparticles: a novel non-radioactive method. PLoS One 2010, 5, e9536, doi:10.1371/journal.pone.0009536. Zheng, B.; von See, M.P.; Yu, E.; Gunel, B.; Lu, K.; Vazin, T.; Schaffer, D.V.; Goodwill, P.W.; Conolly, S.M. Quantitative Magnetic Particle Imaging Monitors the Transplantation, Biodistribution, and Clearance of Stem Cells In Vivo. Theranostics 2016, 6, 291-301, doi:10.7150/thno.13728. Chu, D.; Gao, J.; Wang, Z. Neutrophil-Mediated Delivery of Therapeutic Nanoparticles across Blood Vessel Barrier for Treatment of Inflammation and Infection. ACS Nano 2015, 9, 11800-11811, doi:10.1021/acsnano.5b05583.

Round 2
Reviewer 2 Report
Nothing to add.
The authors answered to most of my comments, even if no in vivo test nor experiences under inflammatory conditions in vitro were performed.